# White matter integrity in mice requires continuous myelin synthesis at the inner tongue

Martin Meschkat[1,2,3,4,10], Anna M. Steyer [1,2,4,11], Marie-Theres Weil[1,2,4], Kathrin Kusch [1,12], Olaf Jahn[4,5,6], Lars Piepkorn[5,6], Paola Agüi-Gonzalez [7], Nhu Thi Ngoc Phan[7,13], Torben Ruhwedel [1,2], Boguslawa Sadowski[1,2,4], Silvio O. Rizzoli [4,7,8], Hauke B. Werner [1], Hannelore Ehrenreich [4,9], Klaus-Armin Nave [1,4] & Wiebke Möbius [1,2,4,8✉]

Myelin, the electrically insulating sheath on axons, undergoes dynamic changes over time. However, it is composed of proteins with long lifetimes. This raises the question how such a stable structure is renewed. Here, we study the integrity of myelinated tracts after experimentally preventing the formation of new myelin in the CNS of adult mice, using an inducible *Mbp* null allele. Oligodendrocytes survive recombination, continue to express myelin genes, but they fail to maintain compacted myelin sheaths. Using 3D electron microscopy and mass spectrometry imaging we visualize myelin-like membranes failing to incorporate adaxonally, most prominently at juxta-paranodes. Myelinoid body formation indicates degradation of existing myelin at the abaxonal side and the inner tongue of the sheath. Thinning of compact myelin and shortening of internodes result in the loss of about 50% of myelin and axonal pathology within 20 weeks post recombination. In summary, our data suggest that functional axon-myelin units require the continuous incorporation of new myelin membranes.

[1] Department of Neurogenetics, Max Planck Institute of Experimental Medicine, Göttingen, Germany. [2] Electron Microscopy Core Unit, Max Planck Institute of Experimental Medicine, Göttingen, Germany. [3] Göttingen Graduate Center for Neurosciences, Biophysics, and Molecular Biosciences (GGNB), Göttingen, Germany. [4] DFG Research Center for Nanoscale Microscopy and Molecular Physiology of the Brain (CNMPB), Göttingen, Germany. [5] Proteomics Group, Max Planck Institute of Experimental Medicine, Göttingen, Germany. [6] Translational Neuroproteomics Group, Department of Psychiatry and Psychotherapy, University Medical Center Göttingen, Göttingen, Germany. [7] Department of Neuro- and Sensory Physiology, University Medical Center Göttingen, Center for Biostructural Imaging of Neurodegeneration, Göttingen, Germany. [8] Cluster of Excellence "Multiscale Bioimaging: from Molecular Machines to Networks of Excitable Cells" (MBExC), University of Göttingen, Göttingen, Germany. [9] Clinical Neuroscience, Max Planck Institute of Experimental Medicine, Göttingen, Germany. [10] Present address: Abberior Instruments GmbH, Göttingen, Germany. [11] Present address: Imaging Centre, European Molecular Biology Laboratory (EMBL), Heidelberg, Germany. [12] Present address: Institute for Auditory Neuroscience and InnerEarLab, University Medical Center Göttingen, Göttingen, Germany. [13] Present address: Department of Chemistry and Molecular Biology, University of Gothenburg, Gothenburg, Sweden. ✉email: moebius@mpinat.mpg.de

Beyond the fundamental property of increasing conduction velocity at a low energy cost, myelin is involved in the organization and functioning of the brain and is more dynamic than previously anticipated. Most of the myelin of the central nervous system (CNS) is formed by oligodendrocytes during development[1]. Indeed, a study in humans showed that a stable population of mature oligodendrocytes is established in childhood and remains static throughout life[2]. Apart from these early differentiated oligodendrocytes, new oligodendrocytes are continuously generated from oligodendrocyte progenitor cells (OPCs) that persist during adulthood[3–6]. The role of these adult-formed oligodendrocytes in myelin plasticity during motor learning and their relevance for replacement of aged oligodendrocytes and their myelin sheaths is the subject of intense investigation[5,7,8]. As the early differentiated oligodendrocytes are long lived and survive alongside the newly formed cells, it is unlikely that the principal role of adult-born oligodendrocytes is the replacement and turnover of the developmentally formed myelin sheaths[9]. Instead, they increase the total number of oligodendrocytes and add additional myelin to the existing white matter[2,9–11].

Myelin synthesized during development is continuously exchanged and renewed. However, as the myelin sheath is a tightly packed plasma membrane extension with limited access from the soma, and since a single oligodendrocyte can maintain many myelin sheaths, it is plausible that its turnover occurs slowly. In a pulse-chase experiment using stable-isotope labeling with $^{15}$N and mass spectrometry, long-lived proteins were identified in the rat brain[12]. Indeed, myelin proteins such as proteolipid protein (PLP) and myelin basic protein (MBP) retained $^{15}$N in 20% of the peptides after a chase period of 6 months[12]. In a recent study of protein lifetimes in the brain using in vivo isotope labeling, myelin proteins appeared in the extremely long-lived population with a half-life between 55 days (2′,3′-cyclic-nucleotide 3′-phosphodiesterase (CNP)) and 133 days (claudin11 (Cldn11))[13]. Further, targeted disruption of the *Plp* gene in the adult showed that PLP has a half-life of ~6 months[14]. These data suggest that the myelin sheath is turned over and renewed by the respective oligodendrocyte in a continuous but protracted process.

Since the renewal of the myelin sheath is not well understood, we directly visualized the turnover of myelin internodes in the adult mouse by ultrastructural analysis. To investigate the maintenance of compact myelin, we generated a mouse line with a floxed exon 1 of the gene encoding MBP, which is common to all classical MBP isoforms[15–17]. MBP is an essential structural component of the CNS myelin, driving the adhesion of the cytosolic membrane leaflets required for the formation of multilayered compact myelin[18–22]. Accordingly, the lack of MBP in the mouse mutant *shiverer* prevents myelin compaction[23–25]. Therefore, oligodendrocyte processes only loosely associate with axons and fail to establish a stable compact myelin sheath. Exploiting this as a distinguishing structural feature, we investigated the long-term stability and apparent half-life of individual compact myelin sheaths. For this purpose, we crossed the MBP-flox line with the oligodendrocyte-specific inducible *Plp*-Cre$^{ERT2}$ driver line[26]. This inducible *Mbp* ablation allowed us to eliminate MBP biosynthesis in mature oligodendrocytes, and to prevent the de novo formation of compact myelin at an age when most developmental myelination has been achieved. After induction, MBP biosynthesis was abolished, resulting in structural changes of the myelin sheath due to the lack of compaction of newly formed myelin membranes. We used mass spectrometry imaging of $^{13}$C-lysine pulse-fed mice by nanoscale secondary ion mass spectrometry (NanoSIMS), a technique to investigate the isotopic composition of samples with high mass sensitivity and lateral resolution[27]. After *Mbp* ablation, unusually enlarged inner tongue structures showed a higher content of $^{13}$C than compact myelin. In the current study, these structural transformations resembling a *shiverer*-like myelin phenotype were investigated in detail by various electron microscopy techniques. Using this model to visualize adult myelin turnover, we identified sites of insertion of newly synthesized myelin-like membranes and manifestations of myelin removal. In this work, the determination and localization of myelin turnover reveal the dynamic nature of maintenance mechanisms and provide insight into the life of a myelin sheath during ageing.

## Results

**Adult MBP ablation induces demyelination with the survival of oligodendrocytes.** To allow inactivation of the *Mbp*-gene in the adult we established a mouse line with a lox-P flanked exon 1 of the classical *Mbp* locus (Mbp$^{fl/fl}$) (Fig. 1A). By interbreeding with mice expressing tamoxifen-inducible Cre$^{ERT2}$ in myelinating cells under the control of the *Plp* promotor[26] we gained control mice (Mbp$^{fl/fl}$*Plp$^{CreERT2−/−}$) and inducible knockout mice (Mbp$^{fl/fl}$*Plp$^{CreERT2+}$). Mice of both genotypes were treated with tamoxifen at the age of 8 weeks (Fig. 1B). Genomic PCR analysis of brain lysate at 6 months after tamoxifen-induction confirmed recombination of the floxed allele only in (Mbp$^{fl/fl}$*Plp$^{CreERT2+}$) mice (Fig. 1C), termed hereafter, inducible conditional knockout mice (iKO) in comparison to tamoxifen-injected (Mbp$^{fl/fl}$) control mice. Time points of analysis are indicated as weeks post tamoxifen induction (pti).

The relative abundance of *Mbp* mRNA in the brain was determined 24 h and 5, 10, and 20 days after the first tamoxifen injection (Fig. 1D). The mRNA level in the brain declined to 50% and 23% after 10 days and 20 days, respectively. As the mutant mice aged, the *Mbp* mRNA abundance in the brain partly recovered to 82% at 1 year pti, suggesting *Mbp* expression by newly differentiated oligodendrocytes derived from non-recombined OPCs (Fig. 1E). The expression levels of mRNAs encoding the myelin proteins PLP, myelin-associated glycoprotein (MAG), myelin oligodendrocyte glycoprotein (MOG), and CNP, were largely unchanged (Fig. S1A). The ablation of exon 1 of the classical *Mbp* gene did not impair the expression of *Golli* mRNA which is encoded by the same transcription unit (Fig. 1A and Fig. S1A).

Since myelin proteins are known to exhibit a long lifetime, we next analyzed the MBP protein abundance in total brain lysate in our mouse model of adult *Mbp* ablation. As expected, MBP levels progressively decreased, showing a significant reduction at 8, 16, and 26 weeks pti (Fig. 1F). From this immunoblot analysis, we calculated an apparent half-life of approximately 77 days (11 weeks). The myelin proteins PLP, MOG, and MAG, also decreased in abundance in brain lysate, but to a lesser extent (Fig. S1B–D). By 26 weeks pti, MBP levels were reduced to 26% of control. At 40 weeks pti, MBP remained reduced to levels comparable to 26 weeks pti, but recovered to control amounts at the latest time point of 52 weeks pti (Fig. 1F). The amount of PLP reached control levels at 40 weeks pti; MOG and MAG levels recovered at 52 weeks pti (Fig. S1D).

To discriminate between the possibilities that MBP is lost from the myelin sheath or that the amount of compact myelin itself is decreased, we purified myelin from brain lysate and analyzed the relative abundance of MBP and PLP. As shown in Fig. 1G and Fig. S2A, at 26 weeks pti, MBP abundance was significantly reduced in brain lysate. In contrast, MBP levels were not changed in a purified myelin fraction, but the myelin fraction of iKO mice was visibly reduced, indicating a loss of compact myelin (Fig. S2B, C).

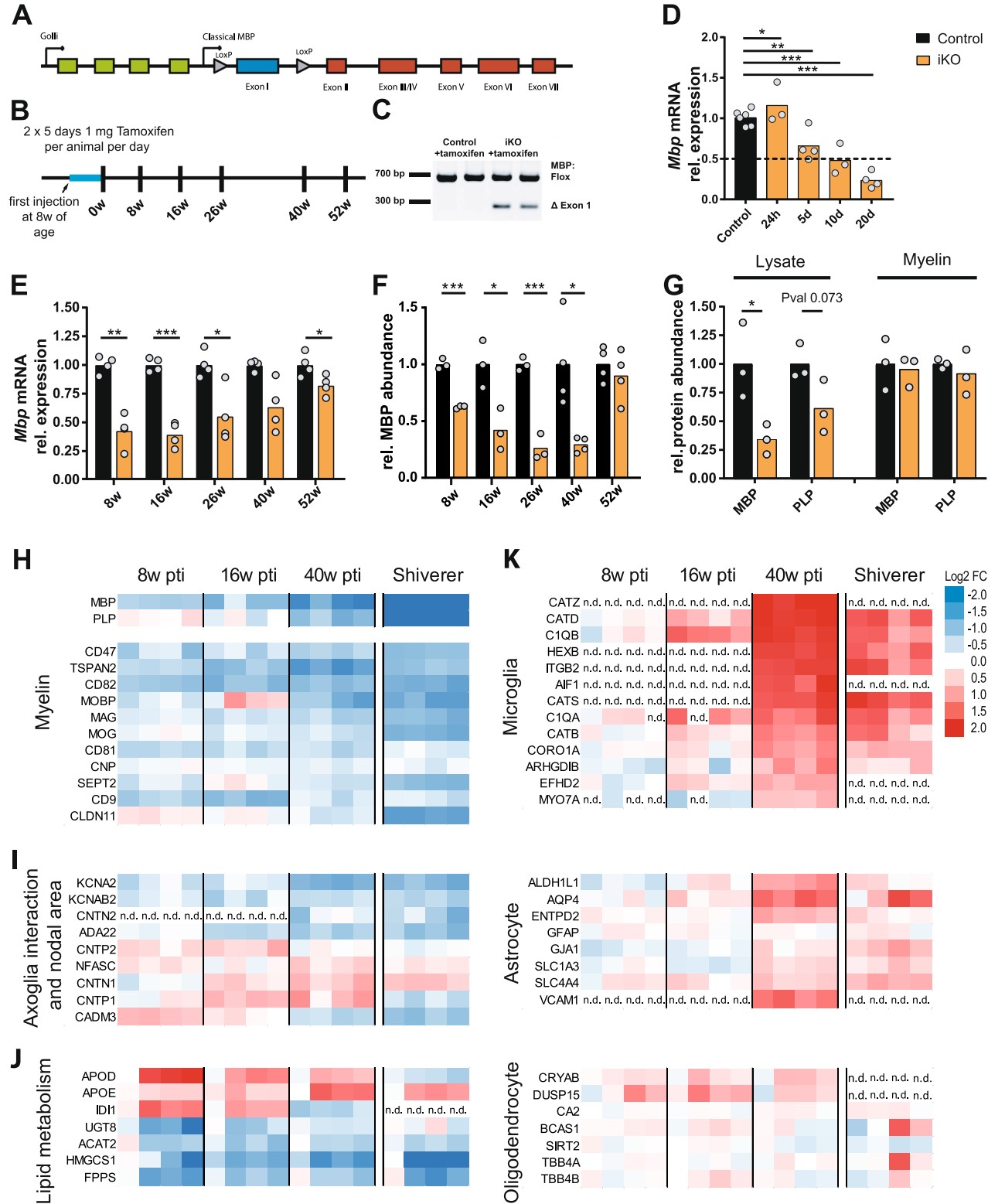

To determine whether this *Mbp*-ablation-induced demyelination was caused by the loss of oligodendrocytes, we investigated oligodendrocyte numbers and the proliferation of OPCs. The determination of OPC and oligodendrocyte numbers by labeling platelet-derived growth factor receptor alpha (PDGFRA) and oligodendrocyte lineage transcription factor 2 (Olig2), respectively (Fig. S3A, B), at 46 weeks pti was performed in the fimbria; a comparatively homogenous white matter tract. We found a significantly increased density of Olig2+ and PDGFRA+ cells in

the iKO, while the area of the fimbria remained unchanged (Fig. S3C). In addition, significantly increased numbers of CAII positive oligodendrocytes in the iKO were found 26, 46 and 52 weeks pti (Fig. S3D). In accordance, TUNEL staining did not indicate increased apoptosis of cells at 40 and 52 weeks pti (Fig. S4A, B). To track proliferating cells, we administered 5-ethynyl-2′-deoxyuridine (EdU) at 40 weeks pti, for 3 weeks, followed by an EdU-free chase period of 3 weeks. Double staining with EdU revealed that the percentage of EdU-positive Olig2+

**Fig. 1 Deletion of the *Mbp* gene in mature oligodendrocytes and subsequent molecular changes. A** Schematic of *Mbp* gene structure with floxed exon 1. **B** Experimental design and time points of analysis. **C** Floxed exon 1 of the classical *Mbp* locus is deleted upon tamoxifen injection using the inducible *Plp*-CreERT2 mouse line. This PCR result was confirmed in $n = 3$ mice (Leone et al.[26]). **D, E** Relative *Mbp* mRNA abundance in brain lysate at the indicated time points post tamoxifen injection (pti) in male mice. The stippled line indicates a reduction to 50%. **D** Two-tailed unpaired *t* test: control vs iKO: 24 h pti: $p = 0,0259$; 5 days pti: $p = 0.035$; 10 days pti: $p = 0.0016$; 20 days pti: $p < 0.0001$. **E** Two-tailed unpaired *t* test: control vs iKO: 8 weeks pti: $p = 0.01972$; 16 weeks pti: $p = 0.00655$; 26 weeks pti: $p = 0.042$; 40 weeks pti: $p = 0.0598$; 52 weeks pti: $p = 0.0373$. **F** Relative abundance of MBP in brain lysate at the indicated time points by immunoblot analysis. multiple *t* tests: control vs iKO: 8 weeks pti (female mice): $p = 0.0002$; 16 weeks pti (male and female mice): $p = 0.022$; 26 weeks pti (female mice): $p = 0.0006$; 40 weeks pti (male mice): $p = 0.012$; 52 weeks pti (male mice): $p = 0.47$. **G** Immunoblot analysis of MBP and PLP abundance in lysate and myelin fraction 26 weeks pti in male mice. The protein abundance in the myelin fraction is unchanged (see also Figs. S1 and S2). Two-tailed unpaired *t* test: control vs iKO in brain lysate: MBP abundance: $p = 0.032$; PLP abundance: $p = 0.073$; control vs iKO in myelin purification: MBP abundance: $p = 0.7664$; PLP abundance: $p = 0.5$. ($p < 0.05$ (*), $p < 0.01$ (**), $p < 0.001$ (***)). Single data points in the graphs represent individual mice (*n*-number). Source data are provided with this paper. **H–K** Heatmaps of the normalized abundance of proteins selected from the quantitative proteome analysis of whole optic nerve in iKO at the indicated time points and *shiverer* mice at the age of 10 weeks (all male mice). Myelin proteins (**H**), proteins involved in axo-glia interaction and present in the node area (**I**), and proteins of lipid metabolism (**J**) are depicted. Markers of microglia, astrocytes, and oligodendrocytes (**K**) were assigned to the cell type according to Zhang et al.[28] Shown are the averages of two technical replicates from $N = 4$ mice, optic nerve lysate, 8, 16, and 40 weeks pti and *shiverer* at 10 weeks of age. For normalization, iKO abundance values were divided by the mean of the corresponding control group with the color code representing downregulation (blue) or upregulation (red) as $\log_2$-transformed fold-change. Abundance values for MBP and PLP were derived from a dataset recorded in the MS$^E$ acquisition mode dedicated to correct quantification of exceptionally abundant proteins (see also Fig. S3–S5, Supplementary Table 1, and "Methods" for details), n.d. not detected.

and PDGFRA$^+$ cells had increased 4-fold in the iKO (Fig. S3E, E′, F, F′). Approximately 3 times more PDGFRA$^+$ cells were also EdU positive in the iKO compared to control, (Fig. S3F, F′). However, within the period of the EdU administration and chase these OPCs did not differentiate to CAII positive oligodendrocyte in significant numbers (Fig. S3G, G′). Expression analysis in the corpus callosum 40 and 52 weeks pti showed unchanged or increased expression of PLP, Olig2, PDGFRA, and CAII, while MBP expression was significantly reduced (Fig. S4C). We conclude that recombined oligodendrocytes persist and continue cell-type-specific protein expression while newly differentiated oligodendrocytes might partially account for an increase in myelin gene transcripts.

**Proteome analysis shows similarities between *Mbp* iKO and *shiverer* mice.** To systematically characterize changes in the abundance of myelin proteins, we utilized proteome analysis by quantitative mass spectrometry. We chose the optic nerve because of its high degree of myelination and ease of extraction as an intact structure. Moreover, the myelination pattern remains stable once myelination is completed. Therefore, it served as a model tissue for myelin maintenance also in the subsequent ultra-structural investigations.

For proteome analysis, we obtained whole optic nerve lysates from iKO and controls at 8, 16, and 40 weeks pti and for comparison, optic nerve lysates from 10-week-old *shiverer* mice, which fail to form compact myelin due to the lack of MBP[25] (Fig. 1H, K). At this time point, *shiverer* mice reach a clinical end stage. In total, we identified and quantified 1863 proteins with an average sequence coverage of 38.5% from the iKO/control samples at the three time points, and 1690 proteins with an average sequence coverage of 33.8% from the *shiverer*/control samples at 10 weeks (Supplementary Data and Fig. S5). The proteins were assigned to the enriched expression in cell types according to the RNA-Seq transcriptome[28]. Apart from analyzing the protein abundance changes upon MBP deletion for each time point individually (iKO vs Ctrl; Supplementary Data), we also compared early (8 weeks) and late (40 weeks) iKO/Ctrl ratios to detect differences in normalized protein abundance over the course of MBP deficiency (iKO/Ctrl 40w vs iKO/Ctrl 8w; Fig. S5 and Supplementary Data). Guided by this analysis, we selected proteins of interest from the entire iKO/Ctrl dataset and compared their normalized abundance with that in *shiverer* mice, as a proxy for the demyelination endpoint (see heatmaps in

Fig. 1). Indeed, we found numerous myelin proteins reduced in abundance (MBP, PLP, MAG, MOG, CNP, claudin 11 (CLDN11), CD9, and tetraspanin-2 (TSPAN2)) (Fig. 1H), while markers of oligodendrocyte cell bodies were unchanged or slightly elevated (carbonic anhydrase 2 (CAH2), BCAS1, CRYAB) (Fig. 1K). Indicating neuropathology, levels of microglial markers (cathepsins, iba1 (AIF1), HexB, and complement subcomponents, C1QB and C1QA)) and astrocyte markers (ALDH1L1, AQP4) were increased at the late time point in the iKO (Fig. 1K). Virtually complete absence of MBP was confirmed in the *shiverer* optic nerve proteome (Supplementary Data). In addition, we found a strong reduction in the number of other myelin proteins also in *shiverer*, such as PLP, claudin11, septin2, septin4, MAG, and tetraspanin-2. Levels of oligodendrocyte markers SIRT2 and CAH2 were almost unchanged, confirming that MBP is not required for oligodendrocyte survival (Fig. 1K). The abundance of microglial markers was similarly elevated in *shiverer* as in iKO mice 40 weeks pti. In addition, astrocytic markers were increased in *shiverer* indicating astrogliosis. Interestingly, chronic MBP deficiency or induced loss of MBP, 40 weeks pti, resulted in similar changes in the proteome. These included elevated levels of proteins involved in axo-glia interaction at the paranode like contactin-1 (CNTN1) and neurofascin (NFASC), as well as decreased amounts of CADM3 at the internode (Fig. 1I). In the iKO as well as in *shiverer*, the levels of juxtaparanodal voltage-gated potassium channel α subunit K$_v$1.2 (KCNA2) and the subunit K$_v$β2 (KCNAB2) were also decreased. Furthermore, we found significantly diminished levels of their interaction partner ADAM22, as well as contactin-2 (CNTN2), indicating alterations in the juxtaparanodal organization.

As depicted in Fig. 1J we also detected abundance changes in proteins involved in lipid metabolism. These comprise enzymes of the isoprenoid and cholesterol biosynthetic pathway such as HMG-CoA-synthase (HMGCS1), isopentenyl-diphosphate delta-isomerase 1 (IDI1), and farnesyl pyrophosphate synthase (FDPS). HMG-CoA-synthase 1, which catalyzes the rate-limiting step in this pathway, was reduced in iKO as well as in *shiverer*. In addition, levels of apolipoproteins, Apo D (APOD, expressed in oligodendrocytes) and Apo E (APOE, expressed in microglia and astrocytes), involved in cholesterol transport, were increased in the iKO. Another enzyme important for myelin lipid synthesis, UDP-galactose:ceramide galactosyl-transferase (UGT8), was reduced in abundance. Taken together, after induced ablation of MBP, the subsequent loss of myelin proteins was accompanied

by changes in proteins involved in axo-myelinic interaction and myelin lipid synthesis. Similarities in the whole optic nerve proteome of *shiverer* and iKO mice suggest that progressive demyelination by the deletion of MBP in adulthood induced a state that resembles, in several aspects, the dysmyelinated situation in *shiverer*. Importantly, however, the *Mbp* iKO mouse allows visualization of the morphological changes occurring during demyelination.

To validate the indications of neuropathology found by proteome analysis, we used immunohistochemistry for GFAP, MAC3, APP, and CD3 in the fimbria (Fig. S6). An increase in the GFAP-immunopositive area was detected 16 weeks pti, together with a significant increase in the number of CD3 immunopositive cells. This was followed by increased MAC3-immunopositive area and the appearance of APP-positive axonal spheroids at 26 weeks pti. These signs of neuropathology were progressive. In conclusion, the ablation of MBP in mature oligodendrocytes caused a slowly progressing demyelination without impairment of myelin gene expression or oligodendrocyte survival, accompanied by a slowly developing neuropathology, validating our mouse model as suitable for a fine structural analysis of the maintenance of the myelin sheath.

**Metabolic stable-isotope labeling of iKO mice and NanoSIMS.** Next, we used the *Mbp* iKO model to directly visualize the integration of newly synthesized proteins into the mature myelin sheath using NanoSIMS. For this purpose, we applied stable isotope labeling by pulse-labeling iKO mice for 45 days with a $^{13}C$-lysine diet starting 18 weeks pti, according to ref. [13], followed by 1 week of chase with a normal diet. Tissue was collected at 26 weeks pti. Considering the lateral resolution of NanoSIMS we used spinal cord samples which contain large myelinated fibers. The samples were prepared for transmission electron microscopy and mapped for the occurrence of phenotypic changes of the compact myelin structure, due to the lack of MBP. The most striking feature in comparison to the control was tubular-vesicular enlargements of the inner tongue in the iKO sample (Fig. 2A). The EM images were correlated with their respective $^{12}C^{14}N$ and $^{13}C^{14}N$ ion images. By manually selecting small regions of interest, the local isotopic ratio of $^{13}C^{14}N/^{12}C^{14}N$ was determined on several morphological categories as described in Fig. 2. In detail, we assessed the $^{13}C^{14}N/^{12}C^{14}N$ ratio on structures in the enlarged inner tongue (Fig. 2A, C), on compact myelin, myelin debris, and a myelinoid body (Fig. 2B, C). From the statistical analysis, compact myelin structures showed significantly less enrichment of $^{13}C$ than the axon or the structures in the enlarged inner tongue (Fig. 2C). This indicates that proteins in the axon and the inner tongue structures were turned over faster compared to the compact myelin sheath. Similar results were obtained from an iKO mouse pulse-labeled for 60 days (Fig. S7).

**Shiverer-like membranes replace compact myelin.** To investigate the conspicuous structures at the inner tongue in detail, we assessed this phenotype in iKO mice by ultrastructural analysis of the optic nerve. We detected structural phenotypes described below, not only in the optic nerve but also in the spinal cord and corpus callosum with differences in extent and onset. Subsequently, we investigated the optic nerve in detail due to its high degree of myelination and stable myelination pattern.

We observed demyelination at 16 weeks pti that became widespread at 26 weeks pti (Fig. 3A). Alongside demyelinated axons, membrane processes emerged that resembled the *shiverer* phenotype. These were most obvious at the myelin inner tongue or both at the inner and outer tongue. Only occasionally,

membrane tubules could be found adjacent to non-myelinated axons (Fig. 3B, C). Such *shiverer*-like membrane tubules often left behind residual myelin sheaths that incompletely covered the axon (lower left panel in Fig. 3B). We quantified the occurrence of myelin tubulations at 16 weeks pti (Fig. 3C). Outer tongue tubulations were only clearly identified where inner and outer tongue tubulations occurred at the same myelinated axon (Fig. 3B, upper right panel). Tubules in close vicinity to non-myelinated axons were considered as "adjacent to axons" (Fig. 3B lower right panel).

Quantification showed that the loss of compact myelin involved a transition phase characterized by the appearance of these membrane tubules resembling *shiverer*-like myelin membranes ("pathological appearing myelinated axons" in Fig. 4A). At 52 weeks pti 70% of all axons were demyelinated (Fig. 4A). Interestingly, we did not observe axonal loss. This might be explained by the very slow demyelination without oligodendrocyte loss that allowed for an axonal adaptation as described in *Plp1* transgenic mice[29].

We next asked whether the dynamics of myelin renewal are influenced by ageing. For this purpose, we induced *Mbp* ablation at the age of 6 months and analyzed the mice 26 weeks pti (Fig. 4B). We observed similar phenotypic changes including tubulation (pathological appearing myelinated axons) and demyelination, but to a lesser extent. This indicates an age-related slowing down of myelin renewal.

To determine possible indicators of remyelination, we analyzed the optic nerve 52 weeks after induction at the age of 8 weeks by assessing the number of myelinated axons and myelin thickness in the distal and proximal part of the optic nerve (Fig. S8). We found more myelinated axons compared to the time point 40 weeks pti and more in the proximal part compared to the distal part at 52 weeks pti (Fig. S8A, B). The myelin sheath thickness was reduced compared to control in both regions with thicker sheaths in the proximal region (Fig. S8C, D). These observations suggest some remyelination is taking place at these time points.

To further characterize the *shiverer*-like membrane tubules, we investigated optic nerve cryosections 26 weeks pti (Fig. S9) by immunoelectron microscopy. We found that *shiverer*-like membranes were devoid of MBP but labeled for the major compact myelin protein, PLP. Interestingly, the density of labeling of MBP on compact myelin profiles was similar in control and iKO, but the compact myelin surface area was significantly smaller in the iKO, 26 weeks pti (Fig. S9C–F). This agrees with the biochemical composition of the isolated myelin fraction, as shown above.

To determine the thinning of the compact myelin sheath, we measured the area of the myelinated fiber and the area covered by the tubulated inner tongue and the axon (Fig. 4C). The ratio of the calculated axonal diameter divided by the calculated myelinated fiber diameter after removing the inner tongue area, which we called "corrected *g*-ratio", was then plotted against the axon caliber (Fig. 4D). This corrected *g*-ratio is a measure for the thickness of the remaining compact myelin considering the enlarged tubulated inner tongue area in the iKO (Fig. 4C). At 26 weeks pti, the corrected *g*-ratio scatter plot showed an upshift of the cloud that indicates myelin thinning (Fig. 4D). Indeed, we observed significant thinning of the myelin sheath at 8, 16, and 26 weeks pti (Fig. 4E). The amount of non-myelinated axon profiles (Fig. 4A) indicated a loss of myelin also by shortening of internodes. To better understand these changes, we applied 3D visualization by serial block-face imaging using focused ion beam-scanning electron microscopy (FIB-SEM) (Fig. 5A and Supplementary Movie 1). The 3D reconstruction illustrates a shortened internode and a residual patch of compact myelin. By measuring the total length of the myelin covering an individual

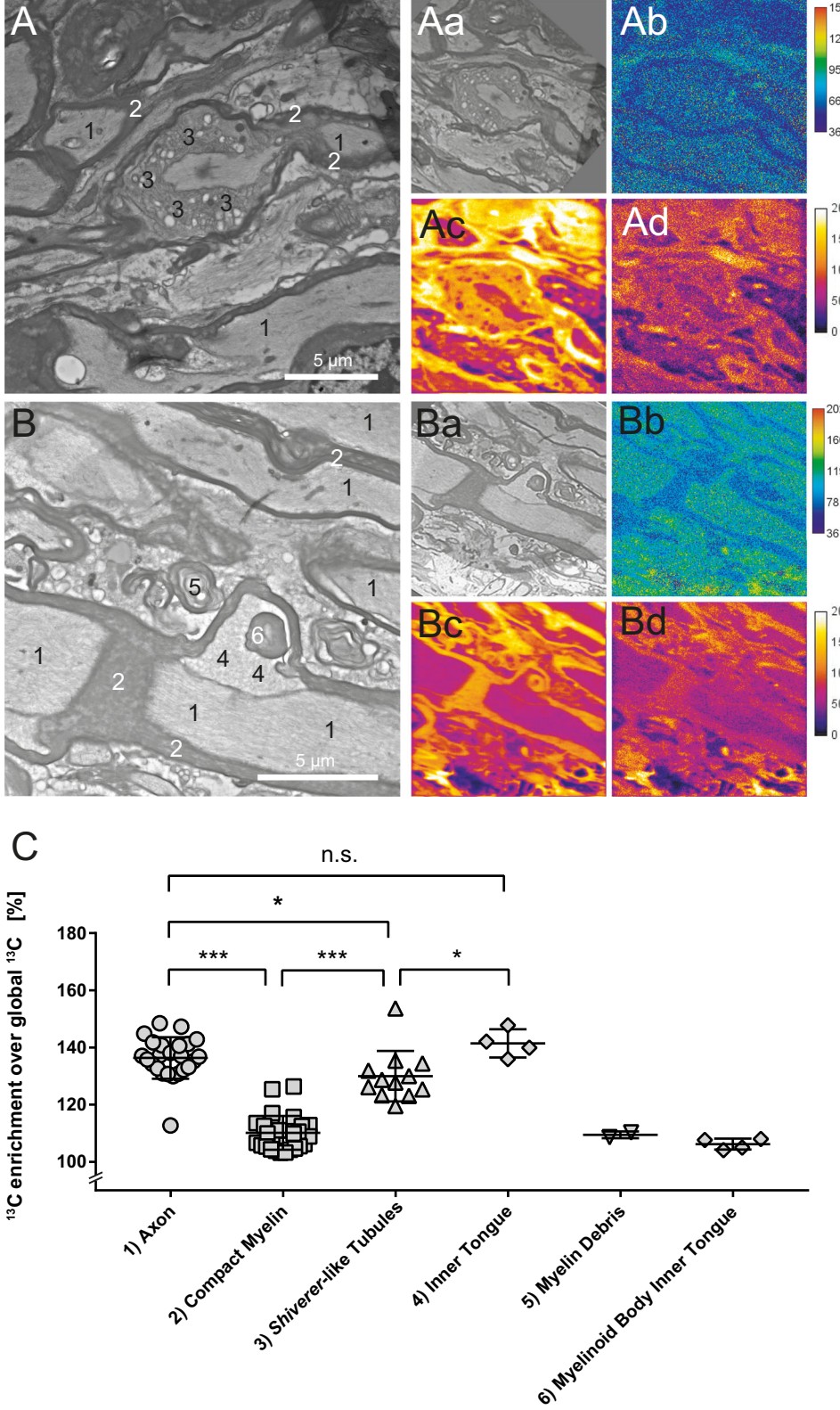

axon within the imaged volume, a myelin coverage could be derived from 3D volumes of both, control and iKO mice at 16 weeks and 26 weeks pti ($n = 3$) (Fig. 5B). This revealed a reduction in myelin coverage by 27% at 16 weeks pti and by 60% at 26 weeks pti. To determine the time course of myelinated internode loss in our model, we counted the number of myelinated axons (normally myelinated and with pathological phenotype) per area at indicated time points and normalized the result to control (Fig. 5C). Since the number of myelinated axons at the time point 8 weeks pti was unchanged compared to control and the reduction of myelinated axons reached the minimum at 26 weeks pti, we determined a 50% loss of myelinated internodes within approximately 20 weeks pti (regression line in Fig. 5C).

**Fig. 2 Determination of myelin turnover by $^{13}$C-lysine feeding and NanoSIMS imaging in *Mbp* iKO.** Longitudinal spinal cord TEM section of iKO ($n = 1$, male) fed for 45 days with $^{13}$C-Lys diet and sacrificed after 1 week of chase with non-labeled control diet at 26 weeks pti. At the indicated structures (numbered 1–6 in **A**–**C**), small regions of interest (ROIs) were sampled manually on the overlaid NanoSIMS isotopic maps and the enrichment of $^{13}$C was calculated from the $^{13}C^{14}N/^{12}C^{14}N$ isotopic ratio of every pixel in the ROIs. **A** Tubular–vesicular enlargement of the inner tongue. **B** A myelinoid body is visible in the inner tongue. **Aa**, **Ba** Aligned TEM image; **Ab**, **Bb** image ratio of $^{13}C^{14}N/^{12}C^{14}N$; **Ac**, **Bc** $^{12}C^{14}N$ NanoSIMS image; **Ad**, **Bd** $^{13}C^{14}N$ NanoSIMS image. **C** $^{13}$C enrichment of the sampled ROI. Every data point (representing the $n$—number used for statistical analysis) corresponds to the average value of the sampled ROI drawn manually on the analyzed structure indicated in **A**, **B**. Numbers on the TEM image correspond to the sampled structures in **C**. Two-tailed unpaired $t$ test: axon vs compact myelin: $p < 0.0001$; axon vs shiverer-like tubules $p = 0.0262$; axon vs inner tongue $p = 0.1933$; compact myelin vs shiverer-like tubules: $p < 0.0001$; shiverer-like tubules vs inner tongue: $p = 0.0287$; (mean $+/-$ SD, $p < 0.05$ (*), $p < 0.001$ (***)). Source data are provided with this paper. Scale bars: **A**, **B** 5 μm.

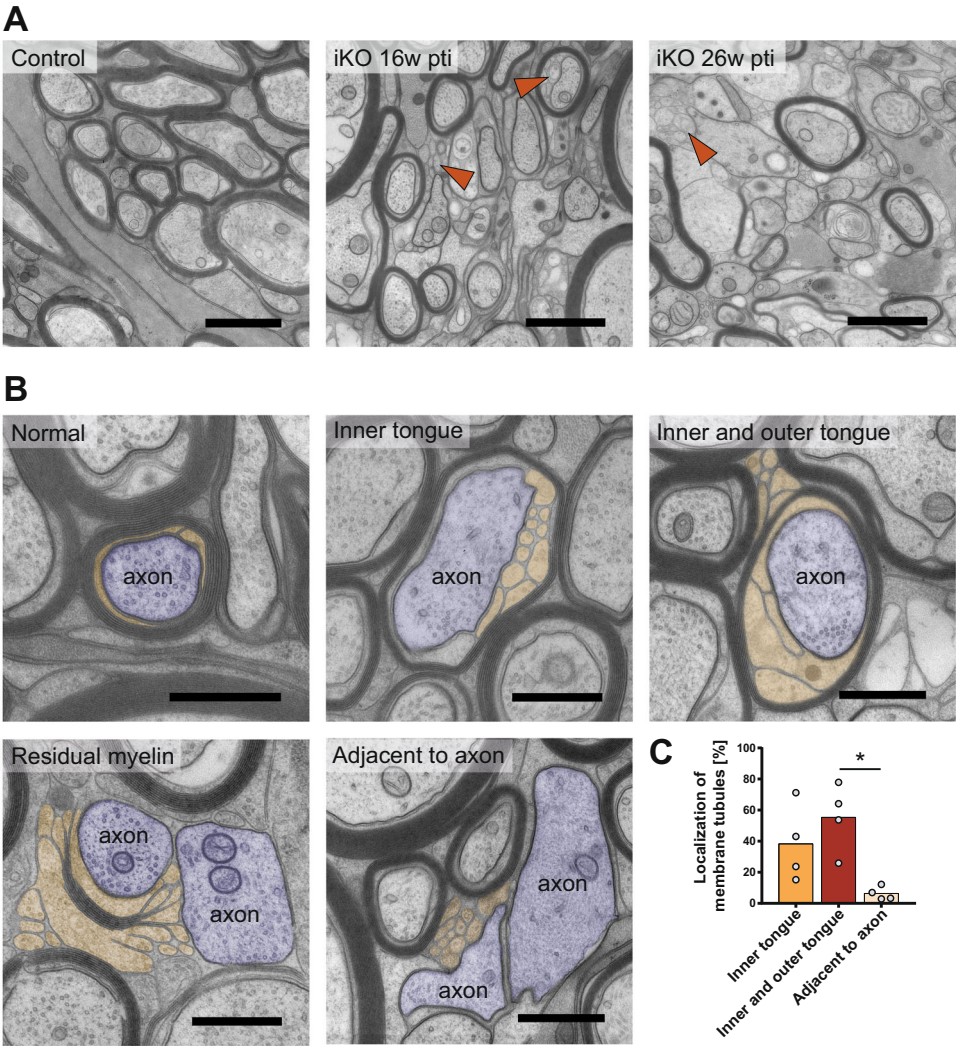

**Fig. 3 Demyelination and emergence of *shiverer*-like membrane tubules in *Mbp* iKO mice. A** Electron micrographs of high-pressure frozen optic nerve showing progressive demyelination. Arrowheads indicate shiverer-like tubules. **B** Illustration of myelin pathology: membrane tubules (colored in orange) emerge at the inner tongue of iKO myelin. Tubulations at the outer tongue of a myelinated axon (colored in purple) are found associated with tubulations also at the inner tongue. At places where most compact myelin is lost, membrane tubules loop out and leave a partially demyelinated axon behind. Tubules are also found next to demyelinated axons. **C** Quantification of the occurrence of membrane tubules. Preparation by high-pressure freezing (HPF) and freeze substitution (FS), optic nerve 16 weeks pti, $n = 4$ iKO animals, 4 random sampled micrographs covering in total 1.600 μm$^2$ were used for quantification (one-way Anova with Tukey's multiple comparison test: inner tongue vs adjacent to axon: $p = 0.1015$; inner tongue vs inner and outer tongue: $p = 0.4545$; adjacent to axon vs inner and outer tongue: $p = 0.0146$ (male and female mice), $p < 0.05$ (*). Source data are provided with this paper. Scale bars **A** 1 μm; **B** 500 nm.

Importantly, 3D visualization revealed that the *shiverer*-like membrane processes are indeed membrane tubules and emerge at the inner tongue (Fig. 5Aa, Ac and Supplementary Movie 1) and also occur at the juxtaparanode and the paranode (Fig. 6A, B and Supplementary Movie 2). The remaining myelin sheaths appear fragmented and transformed into myelin tubules leaving behind some residual patches of compact myelin (Fig. 5Af). In addition to the detachment of paranodal loops, membrane tubules emerged at the paranode and often lost contact to the axon as shown in Fig. 5A, as segmented structures in yellow. For

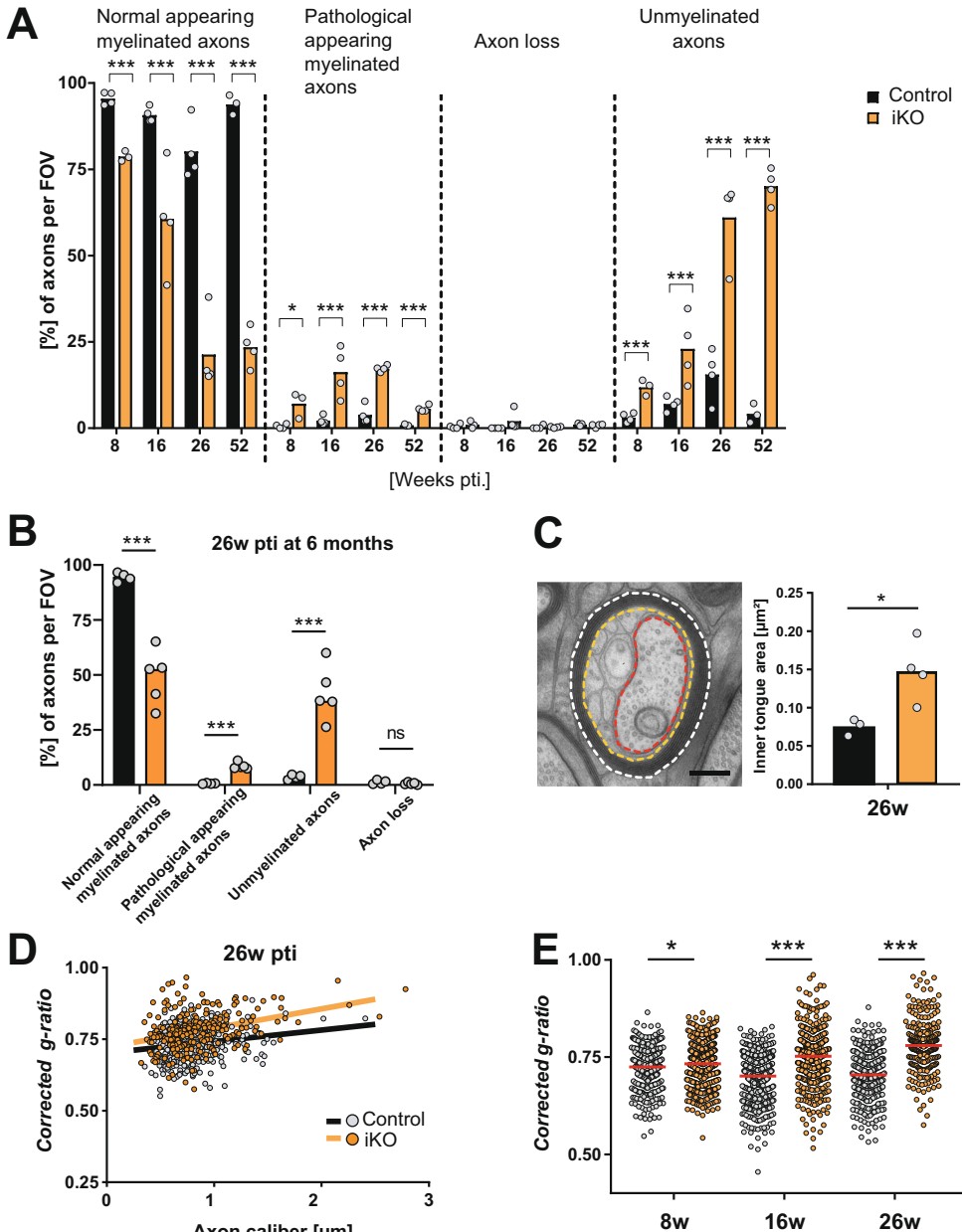

**Fig. 4 Thinning and loss of compact myelin after *Mbp* ablation. A** Quantification of phenotypes at the indicated time points. Analysis was performed on optic nerve cross-sections on a total area of >330 μm² with >200 axons per animal, all axons in the field of view (FOV) were counted. Single data points in graphs represent individual mice (n-number) (two-tailed unpaired *t* test, $p < 0.05$ (*), $p < 0.001$ (***)). Exact *p* values are stated in the respective Source data file. **B** Quantification of phenotypes 26 weeks pti in optic nerve of mice induced at an old age of 6 months (multiple unpaired *t* test, control vs iKO (all male mice)): normal appearing myelinated axons: $p = 0.0002$; pathological appearing myelinated axons: $p < 0.0001$; unmyelinated axons: 0.00053; axon loss: $p = 0.222$; $p < 0.001$ (***). **C** Illustration of corrected *g*-ratio measurement. Three lines are drawn for the area measurement: the outline of the fiber (stippled white line), an outline of the inner border of the compact myelin (orange), and the axon (red). Scale bar: 200 nm. The area of the inner tongue was subtracted from the total fiber area before the calculation of the diameter (see "Methods"). Inner tongue area is increased in iKO ($n = 4$) compared to control ($n = 3$) in optic nerve 26 weeks pti. Measured on TEM cross-sections, at least 150 axons per mouse were analyzed. Unpaired two-tailed *t* test; $p = 0.0294$ ($p < 0.05$ (*)). **D** Scatterplot depicting the corrected *g*-ratios at 26 weeks pti, 150 axons per mouse were analyzed. **E** Corrected *g*-ratio measurements reveal a progressive decrease in compact myelin at the indicated time points, axon calibers pooled (Kolmogorow–Smirnow Test; 8 weeks pti: $p = 0.0155$; 16 weeks pti: $p < 0.0001$; 26 weeks pti: $p < 0.0001$; ($p < 0.05$ (*), $p < 0.001$ (***)) with an n-number of 3 mice for each genotype except at 26 weeks pti with 4 cKO animals. Source data are provided with this paper.

comparison, we analyzed a FIB-SEM data stack from *shiverer* optic nerve at the age of 10 weeks when the animals reach the clinical end stage (Supplementary movie 3). Indeed, the arrangement of tubular oligodendrocyte processes resembled the myelin tubules observed in the iKO. This finding supports our concept, that the induced *Mbp* knockout is gradually

transforming the normal myelin sheath into *shiverer* myelin tubules by integration of newly synthesized material.

To assess the nodal phenotype, we quantified nodes of Ranvier on longitudinal optic nerve cryosections after immunofluorescent staining of Caspr1 and NaV1.6 (Fig. 6C). Already at 16 weeks pti a significant loss of intact nodes was detectable which was even

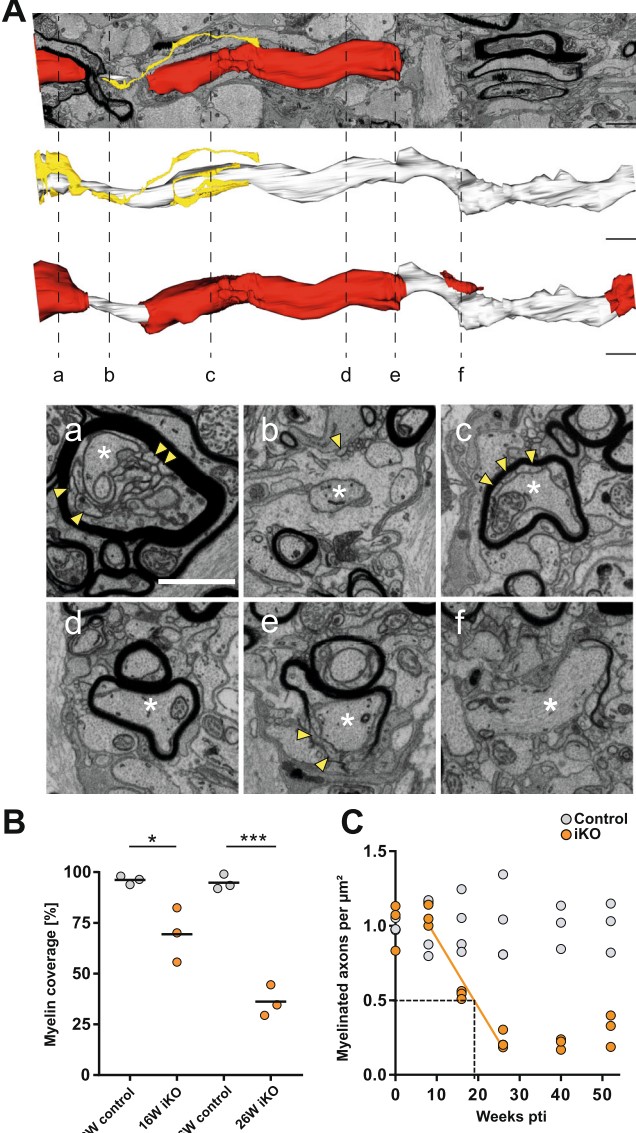

**Fig. 5 Demyelination by internode shortening in *Mbp* iKO mice. A** 3D reconstruction of an image stack acquired by focused ion beam-scanning electron microscopy (FIB-SEM) at 26 weeks pti in the optic nerve of an iKO mouse (shown in Supplementary Movie 1): Yellow: non-compact myelin tubules, white: axon, (white asterisk) red: myelin. At stippled lines, the indicated corresponding image from the stack is shown. Yellow arrowheads point at myelin tubules. Internodes are shortened and fragmented. **B** Quantification of myelin coverage on 3D volumes of iKO and control mice (*n* = 3) 16 weeks pti and (*n* = 3) 26 weeks pti with >90 axons per mouse in the percentage of axonal length within the FIB-SEM volume that is myelinated. Unpaired two-tailed *t* test (16 weeks pti *p* = 0.027; *p* < 0.05 (*); 26 weeks pti *p* = 0.0003; *p* < 0.001 (***), male and female mice). **C** Myelinated axons counted per area on TEM images and normalized to control (*n* = 3). The regression line indicates a 50% loss of myelinated axons within 19–20 weeks pti. At 26 weeks pti, demyelination is maximal. Source data are provided with this paper. Scale bars: **A** 2 µm; **a** 1 µm.

more pronounced at the late time point of 40 weeks pti. The loss of compact myelin in iKO mice and the accumulation of myelin tubules affected the paranodal integrity and the nodal organization. Changes in the abundance of nodal proteins and others involved in axon-glia interaction, as detected by the proteome analysis, support this evidence of disturbed nodal organization (Fig. 1I). We conclude that maintenance of myelin compaction by

continuous MBP synthesis is essential also for paranodal maintenance and integrity of the node.

**Myelin outfoldings, local myelin thinning, and myelinoid bodies**. Maintenance requires a balanced input and output. We asked which indications of myelin disposal are detectable and whether these occur at specific sites. Indeed, we could also observe evidence of myelin removal in the 3D data sets obtained by FIB-SEM in optic nerve samples. Redundant myelin occurs in outfoldings in the form of large sheets of myelin extending into the vicinity of, and often wrapping around neighboring axons (Fig. S10A, A′). Some of the smaller outfoldings appeared as protrusions into adjacent astrocytes (Fig. S10B, B′). Microglia are involved in the clearing of myelin debris in demyelinating conditions as well as recycling of aged myelin[10,30]. As expected, we observed microglia containing typical lysosomes which occasionally appeared close to myelin protrusions (Fig. S10C, C′).

As shown in Fig. 4, the average myelin thickness decreased progressively after induced *Mbp* ablation. In the 3D FIB-SEM data stacks, we observed that within one internode the same myelin sheath varied remarkably in thickness (Supplementary Movie 4). Moreover, we found myelinoid bodies budding at the abaxonal myelin and others protruding into the inner tongue (Fig. 7A, B and Supplementary Movie 4) and also in between myelin lamellae. Myelinoid body formation at the inner tongue could indicate local myelin breakdown and uptake by the oligodendrocyte for reutilization. As we know from the NanoSIMS analysis, these structures showed a similar $^{13}C^{14}N$ to $^{12}C^{14}N$ ratio like compact myelin. We quantified the occurrence of myelinoid bodies in FIB-SEM data stacks in control and iKO mice (*n* = 3) over 10 µm axonal length and found a significant increase in the iKO at 16 weeks pti (Fig. 7C).

Based on our results we propose a mechanism of myelin turnover and renewal by which newly synthesized myelin membrane is incorporated into the sheath predominantly in the adaxonal, non-compact myelin compartment at the inner tongue. Furthermore, we provide evidence that myelin removal is not only mediated by microglia and astrocytes but could also involve oligodendrocyte-intrinsic mechanisms by myelinoid body formation at the inner tongue.

## Discussion

It has been shown recently that mature oligodendrocytes persist life-long in mice as well as humans[2,9] and that their myelin sheaths, which form internodes with the axon, are remarkably stable once formed, with little fluctuation in length[10]. Accordingly, myelin proteins are characterized by slow turnover, with lifetimes in the range of weeks or months[12,13]. Such biochemical studies have revealed the turnover rates of the various protein and lipid components of the myelin sheath. Using inducible ablation of *Plp* in adult mice, it was determined that PLP is reduced by 50% within 6 months[14]. One major obstacle for studies of myelin renewal is the characteristic of the myelin sheath to exclude proteins with bulky fluorescent tags[31]. Therefore, we adopted a different strategy to make the newly synthesized myelin membrane visible, by exploiting the requirement of MBP or myelin compaction. We addressed two related questions: What are the morphological consequences of preventing the replenishment of myelin and what is the time-course of the emerging demyelination. Besides being able to identify new myelin by the non-compacted appearance of *shiverer*-like membranes, the deletion of *Mbp* in the adult mouse by targeting mature oligodendrocytes provided several major findings described below.

First, the reduction in MBP protein levels in the brain to 50% within 77 days (11 weeks) as calculated from our immunoblot

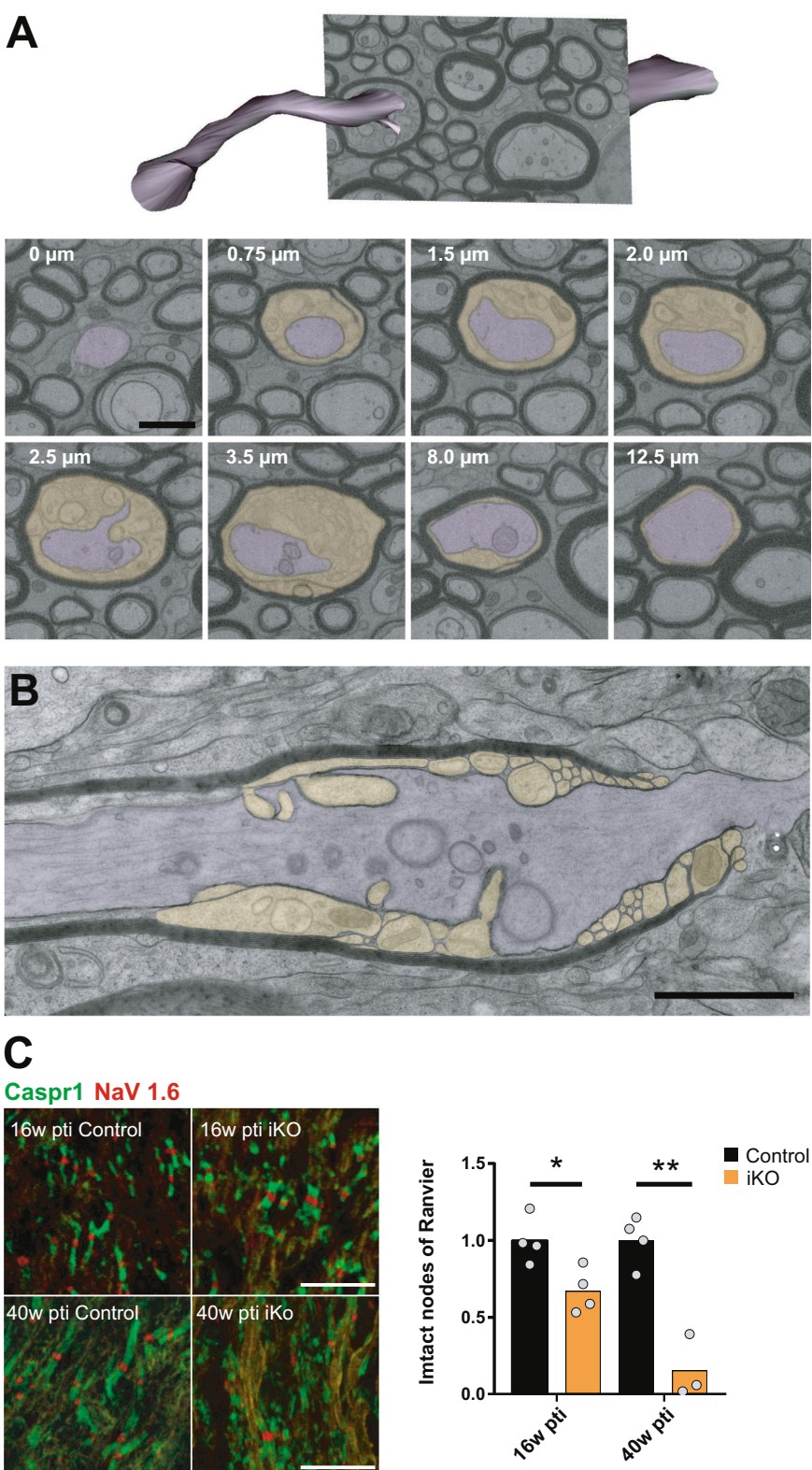

**Fig. 6 Juxtaparanodal myelin tubulation and loss of nodal organization. A** Segmentation of axon and myelin tubules in an image stack acquired by FIB-SEM in the optic nerve of iKO 26 weeks pti (shown in Supplementary Movie 2). The distance along the internode is indicated in the images. Membrane tubules emerge at the juxtaparanode (0.75–3.5 μm) while most of the internode is unaffected. **B** Longitudinal TEM section reveals the juxtaparanode localization of the tubules and the detachment of the paranodal loops. This phenotype was observed independently in two groups of female mice ($n = 7$ controls; $n = 10$ iKOs and $n = 3$ controls; $n = 5$ iKO) and two groups of male mice ($n = 3$ controls; $n = 5$ iKOs and $n = 2$ controls; $n = 4$ iKOs). **C** Confocal light microscopy of immunofluorescence staining of the nodal marker NaV1.6 and paranodal marker Caspr1 on optic nerve cryosections reveals loss of functional nodes of Ranvier (two-tailed unpaired $t$ test, control vs iKO; 16 weeks pti: $p = 0.0207$; 40 weeks pti: $p = 0.0051$ ($p < 0.05$ (*), $p < 0.01$ (**)). Source data are provided with this paper. Scale bars: 500 nm (**A**, **B**), 10 μm (**C**).

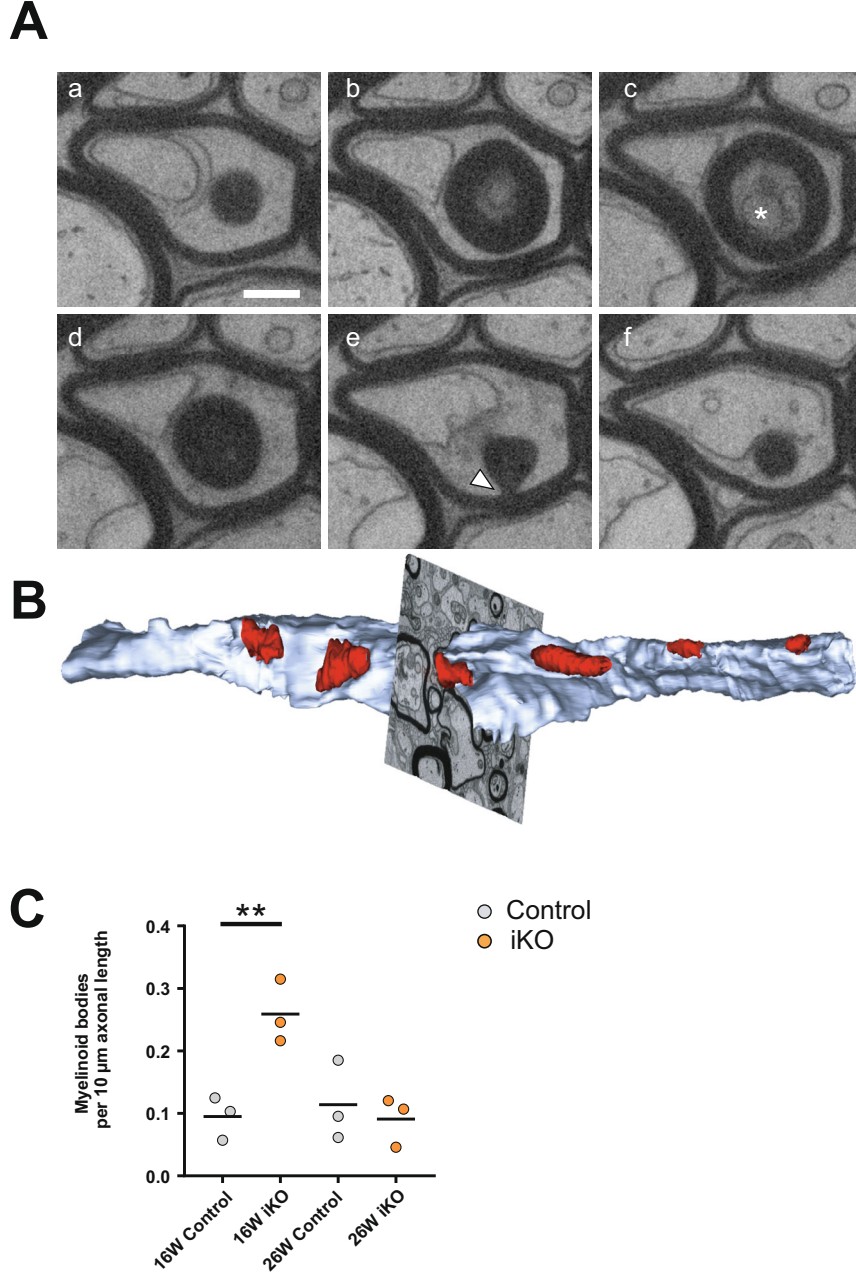

**Fig. 7 Myelinoid bodies at the inner tongue. A** Electron micrographs selected from a FIB-SEM image stack reveal the presence of a myelinoid body (asterisk) at the inner tongue 16 weeks pti in high-pressure frozen optic nerve. This myelinoid body is connected to the myelin sheath (indicated by arrowhead). This observation was made in a group of male mice ($n = 4$ for both genotypes) and a group of female mice ($n = 3$ for both genotypes). **B** Segmentation of myelinoid bodies at the inner tongue from a FIB-SEM image stack (shown in Supplementary Movie 4) acquired 26 weeks pti (blue: axon, red: myelinoid bodies). **C** The number of myelinoid bodies per μm axonal length is increased 16 weeks pti and not significantly different 26 weeks pti. Quantification on 3D volumes with >90 axons per mouse ($n = 3$) (two-tailed unpaired $t$ test, control vs iKO: 16 weeks pti: $p = 0.0097$; 26 weeks pti: $p = 0.1838$: $p < 0.01$ (\*\*)). Source data are provided with this paper. Scale bar: 500 nm.

analysis, matches the measured half-replacement time of 70 days in the cortex[13], indicating normal MBP turnover under these knockout conditions. However, the observed changes in the abundance of other myelin proteins like PLP, MOG, and MAG are likely due to the remodeling of the myelin sheath in this model rather than reflecting their normal half-lives. Second, MBP showed little lateral mobility, evidenced by control levels of MBP in the residual myelin fraction 26 weeks pti and unchanged labeling density in immunoelectron microscopy of remaining compact myelin in the iKO. Third, as already known from

*shiverer* mice, oligodendrocytes differentiate in the absence of MBP expression[32]. Here we showed that adult loss of MBP also does not impair the long-term survival of oligodendrocytes. This was the key minimal requirement for this study of internode turnover in vivo, as the depletion of MBP after a functional myelin sheath had been generated, allowed us to detect *shiverer*-like membranes as indicators of newly synthesized myelin membrane.

We included in the study optic nerves from 10-week-old *shiverer* mice to better understand the observed changes in the iKO,

at both structural and proteomic levels. As shown previously, in *shiverer* mouse, axons are wrapped by fine tubular oligodendrocytic processes that often terminate in loops and also meander among axons without forming sheaths[25]. We concluded that the lack of continuous *Mbp* expression in the iKO mice indeed results in a slow transformation from the previously fully myelinated nerve into a *shiverer* phenotype by replacement of the myelin membranes since we observed similar structures in the iKO.

We identified the inner tongue and more specifically the juxtaparanode as a metabolically active site where newly made components are integrated into the myelin sheath. The inner tongue was already determined as the growth zone in developmental myelination[33,34]. As supported by our NanoSIMS experiment mapping isotope distributions, these cytoplasm-rich compartments of the myelin sheath are the most likely places where newly synthesized components enter the myelin sheath also in adulthood. We note that we cannot formally exclude that the outer tongue also plays an active role in myelin renewal. In our mouse model, distinguishing tubulated left-overs of diminished internodes from potentially tubulated outer tongues was challenging. Since inner and outer tongue tubulations were mostly found in the same myelin sheath, we conclude that the outer tongue tubules occurred closer to the end of the internode, probably representing retracted, loose paranodal loops.

The internodal shortening by the integration of non-compact *shiverer*-membranes at the paranode and juxtaparanode resulted in membrane tubulation and an impairment of the nodal organization. More specifically, in our proteome analysis, we found a concomitant reduction of the protein levels of the juxtaparanodal voltage-gated potassium channel α subunit $K_v1.2$ and the associated disintegrin and metalloprotease ADAM22. This enzyme is an axonal component of the juxtaparanodal macromolecular complex composed of $K_v1.2$[35] and cell adhesion proteins Caspr2 and contactin-2 (TAG-1)[36–38]. In accordance, we found a reduction of contactin-2 abundance at the late time point (40 weeks iKO), and also in *shiverer*. Alterations of $K_v1.1$ and $K_v1.2$ channel subunit distribution were previously described in *shiverer* mice[39]. However, in the Sinha study, an increased abundance of $K_v1.2$ was found in *shiverer* spinal cord whereas we found a decrease in the whole optic nerve proteome. Alterations of paranodal axo-glial junctions, the organization of the juxtaparanode, and retraction of paranodes have been reported in the EAE model of demyelination and in MS patients[40–42]. In contrast to these models of inflammatory demyelination, retraction of paranodes and demyelination developed very slowly in our model, without causing oligodendrocyte loss. Paranodal abnormalities without oligodendrocyte loss also occur in mice with genetic ablation of myelin galactosphingolipid synthesis[43,44]. However, these mice suffer from dysmyelination, develop progressive neuropathology, and die prematurely. In our model, the reduction in the abundance of the UDP-galactose:ceramide galactosyl-transferase (UGT8) and of several enzymes involved in isoprenoid and cholesterol synthesis, is consistent with the observed nodal phenotype. The node of Ranvier maintenance depends not only on the complete set of axo-glial interacting proteins, but also requires intact membrane microdomains composed of galactospingolipids, gangliosides, and cholesterol[45–48]. MBP interacts with negatively charged lipids such as $PI(4,5)P_2$ and thereby influences lipid ordering and associates with galactosylceramide and cholesterol-rich lipid rafts in mature myelin[49–53]. The decrease of MBP levels in the iKO, together with changes in posttranslational modifications of the remaining MBP, e.g. deamination, could affect membrane organization and function[54]. In addition, the breakdown of diffusion barriers by the loss of myelin compaction in our *Mbp* iKO model could also impact paranodal and juxtaparanodal integrity.

Striking morphological similarities exist between our model and the transcription factor *Nkx6-2* null mouse[55] in the form of myelin tubules ('vermicular-like processes') at the inner tongue and compact myelin 'flaps' or stacks flanked by detached paranodal loops. However, these similar phenotypes seem to arise by different mechanisms. In the case of the *Nkx6-2* null mouse, dysregulated expression of paranodal proteins and defects in cytoskeletal remodeling might play important roles, whereas we explain the phenotype in the *Mbp* iKO by the continuous integration of newly formed myelin membrane and the lack of myelin compaction.

We have shown here that induced *Mbp* deletion resulted in a progressive loss of compact myelin. So, how is this compartment eliminated? In accordance with the literature of the time, Hildebrand and colleagues proposed a concept of a metabolically active myelin sheath with lifelong turnover utilizing a "quantal" detachment of Marchi-positive myelinoid bodies as a disposal route[56]. These lamellated myelinoid bodies were found preferentially at the paranodes of large myelinated fibers in the cat spinal cord and inside astrocytes and microglia[57,58]. Different from that, we could not detect a preferential localization of myelinoid bodies. As previously discussed[56], there might be qualitative differences in myelin turnover, regarding myelinoid body formation, between large myelinated fibers like in the spinal cord and small-caliber fibers like in the optic nerve.

As published recently, phagocytosis of myelin debris by microglia increases with age and leads to lipofuscin accumulation in the microglial lysosomal compartment[30]. The same study found similar effects in *shiverer* mice and also described occasional myelin uptake by astrocytes. Age-related accumulation of myelin debris within microglia was also demonstrated by in vivo imaging in the cerebral cortex of the mouse[10]. Myelin phagocytosis by astrocytes seems to play a role in myelin remodeling under certain non-pathological conditions such as internode shortening in the frog optic nerve during metamorphosis[59]. Myelin debris uptake by astrocytes was also described in various forms of white matter injury as an early event in lesion formation leading to enhanced inflammation[60]. Under conditions of myelin degeneration and oligodendrocyte damage, the formation of myelinosomes was identified as an early event in demyelination and lesion development[61]. These myelinosomes show striking similarity in morphology to the myelinoid bodies we found in our model. Yet, in contrast to the study of Romanelli and colleagues (2016), the *Mbp* iKO mouse is unlike any model of acute demyelination because of the absence of oligodendrocyte damage. In our 3D datasets, we also found an increased number of myelinoid bodies inside the inner tongue, resembling a 'myelin inclusion' described in[61]. Such myelinoid bodies were also visible at the inner tongue under control conditions. However, this seems to be an infrequent event that easily escapes electron microscopic observation on thin sections.

A recent study demonstrated that lipid metabolism is essential for myelin integrity[62]. Here, Qki-5, a transcriptional coactivator of the PPARβ-RXRα complex which regulates fatty acid metabolism, was depleted by induced knockout in the adult mouse. Strikingly, rapid and severe demyelination started as early as one week after tamoxifen induction without impairing oligodendrocyte survival. In this model, the loss of myelin lipids was accompanied by conformational changes of MBP promoting dissociation from membranes and loss of myelin compaction. This study indicates that myelin lipids turn over much faster than myelin proteins.

By the application of advanced imaging methods in combination with a novel mouse model to prevent compact myelin

maintenance, we visualized the myelin subcompartment at which newly synthesized myelin membranes are added. We identified the paranodal and juxtaparanodal region at the inner tongue as an important site for myelin renewal in adults and found indications of myelin disposal in form of myelinoid bodies occurring abaxonally and in the inner tongue.

## Methods

**Experimental model and subject details.** All animal experiments were performed in accordance with the German and European animal welfare laws and approved by the Lower Saxony State Office for Consumer Protection and Food Safety (license 33.19-42502-04-16/2119). All mice were housed in standard plastic cages with 1–5 littermates in a 12 h/12 h light/dark cycle (5:30 am/ 5:30 pm) in a temperature-controlled room (~21 °C), with ad libitum access to food and water. All mice used were bred under the C57BL6/N background. Experiments were carried out at the indicated time points after induction with tamoxifen at the age of 8 weeks and in one experiment at 26 weeks mostly in male and sometimes in female mice (indicated in the data). For the generation of the *Mbp*<sup>fl/fl</sup> mouse line, embryonic stem (ES) cells harboring a modified allele of the *Mbp* gene (*Mbp*<sup>tm1a</sup>) carrying a LacZ-neomycin cassette upstream of exon 1 of the classical *Mbp* locus were acquired from the European Conditional Mouse Mutagenesis Program (Eucomm). ES cells were microinjected into blastocysts derived from FVB mice and embryos were transferred to pseudo pregnant foster mothers. For the ES clone B02 germline transmission was achieved by breeding with C57BL/6 N female mice. The resulting offspring harbored the Mbp-lacZ neo allele (*Mbp*<sup>neo/neo</sup>). The construct including the lacZ gene and a neomycin resistance cassette was excised by crossbreeding with mice expressing a FLIP recombinase (129S4/SvJaeSor-Gt(ROSA) 26Sortm1(FLP1)Dym/J; backcrossed to C57BL6/N). Mice expressing tamoxifen-inducible Cre<sup>Ert2</sup> under the control of the *Plp* promotor (MGI:2663093)[26] were obtained from Ueli Suter, ETH Zurich Institute for Molecular Health Sciences, Switzerland. Control mice (Mbp<sup>fl/fl</sup>Plp<sup>CreERT2wt</sup>) and inducible knockout mice (iKO) (Mbp<sup>fl/fl</sup>Plp<sup>CreERT2+</sup>) were generated by breeding Mbp<sup>fl/fl</sup>Plp<sup>CreERT2wt</sup> males with Mbp<sup>fl/fl</sup>Plp<sup>CreERT2+</sup> females. For experiments control and iKO mice were used in groups of littermates of the same sex. Mice were sacrificed at the indicated time points after tamoxifen induction. Genotyping of the *Mbp* flox allele was performed by genomic polymerase chain reaction (PCR) using the following primers:

*Mbp* wt fwd: 5′-GGGTGATAGACTGGAAGGGTTG
*Mbp* wt rev: 3′ of LoxP site: 5′-GCTAACCTGGATTGAGCTTGC
Lar3 rev: 5′-CAACGGGTTCTTCTGTTAGTCC
Genotyping of the Cre<sup>Ert2</sup> allele was performed using the following primers:
5′-CAGGGTGTTATAAGCAATCCC
5′-CCTGGA AAATGCTTCTGTCCG, including a primer pair for CNP as positive control:
5′-GCCTTCAAAC-TGTCCATCTC
5′-CCCAGCCCTTTTATTACCAC

**Tamoxifen induction.** For knock-out induction, 8–9 or 26-week-old mice were injected intraperitoneally with 1 mg tamoxifen (100 µl of 10 mg/ml tamoxifen (Sigma-Aldrich, St. Louis, MO) in corn oil (Sigma-Aldrich)) for 5 consecutive days, followed by a 2-day break and 5 more days of injection as described[26]. To prepare the tamoxifen solution, in a 2 ml tube, 500 µl ethanol and 500 µl corn oil were added to 50 mg tamoxifen and mixed in a tissue lyser (Qiagen, Hilden, Germany) for 10 min at 50 Hertz. After this, the resulting emulsion was added to 4 ml corn oil and mixed until the solution turned clear. The solution was stored in the fridge and used within 5 days. The day after the last tamoxifen injection was considered 0 days pti.

**Expression analyses.** For the characterization of myelin gene expression, RNA from total spinal cord and brain lysates as well as corpus callosum was isolated using QIAzol (Qiagen) and the RNeasy protocols (Qiagen). The concentration and purity of RNA were evaluated using a NanoDrop spectrophotometer (Thermo Fisher Scientific, Waltham, MA, USA). Complementary DNA (cDNA) was synthesized using the Superscript III (Invitrogen, Carlsbad, CA, USA) according to the manufacturer's protocol. Quantitative RT-PCR was performed in triplicates with the GoTaq qPCR Master Mix (Promega) on a LightCycler 480 II PCR system (Roche, Basel, Switzerland). Expression was normalized to the mean of two housekeeping genes Rps13 (Ribosomal Protein S13) and PPIA (Cyclophilin A). Relative changes in gene expression were analyzed using the $2\Delta\Delta C(T)$ method[63]. Primers were designed using the Universal Probe Library from Roche Applied systems (https://www.roche-applied-science.com) and validated using NIH PrimerBlast using Primer3web version 4.1.0 (www.ncbi.nlm.nih.gov/tools/primer-blast/).

Expression of the following genes was Car2 (carbonic anhydrase 2), Cnp, Golli (Golli-MBP), Mag, Mbp, Mog, Olig2, Pdgfa, Plp. All primer sequences are listed in the Table. All primers used for expression analysis were intron-spanning (5′–3′; forward–reverse).

Table. List of primer sequences.

| Primer | Fwd primer [5′–3′] | Rev primer [5′–3′] |
|---|---|---|
| **Housekeeper** | | |
| RPS13 | CGAAAGCACCTTGAGAGGAA | TTCCAATTAGGTGGGAGCAC |
| Ppia | CACAAACGGTTCCCAGTTTT | TTCCCAAAGACCACATGCTT |
| **Myelin genes** | | |
| Car2 | CAAGCACAACGGACCAGA | ATGAGCAGAGGCTGTAGG |
| Cnp | CGCTGGGGCAGAAGAATAC | AAGGCCTTGCCATACGATCT |
| Golli | CCTCAGAGGACAGTGATGTGTTT | AGCCGAGGTCCCATTGTT |
| Mag | AGGATGATGGGGAATACTGGT | AAGGATTATGGGGGCAAACT |
| Mbp | GCCTCCGTAGCCAAATCC | GCCTGTCCCTCAGCAGATT |
| Mog | ACCTGCTTCTTCAGAGACCACT | GGGGTTGACCCAATAGAAGG |
| Olig2 | AGACCGAGCCAACACCAG | AAGCTCTCGAATGATCCTTCTTT |
| Pdgfra | AAGACCTGGGCAAGAGGAAC | GAACCTGTCTCGATGGCACT |
| Plp1 | CTCCAAAAACTACCAGGACTATGAG | AGGGCCCCATAAAGGAAGA |

**Tissue lysis.** For whole-brain lysate, one hemisphere was homogenized in 4 ml modified RIPA buffer using a T-10 basic Ultra-Turrax (IKA, IKA®-Werke GmbH & CO. KG, Staufen Germany) for 20–30 s on speed setting 3–4. The homogenate was then centrifuged at $1000 \times g$ for 10 min at 4 °C and the supernatant was transferred to a new tube. Protein concentration was determined in triplicates using the BCA protein assay kit (Pierce) according to the manufacturer's manual.

**Myelin purification and immunoblotting.** Purification of a lightweight membrane fraction enriched for myelin was performed as previously described[64]. In brief, half brains of three male control and iKO mice each at 26 weeks post tamoxifen were homogenized in 0.32 M sucrose and added gently on top of 0.85 M sucrose solution and centrifuged for 30 min at $75,000 \times g$ at 4 °C. The interphase, containing crudely purified myelin, was carefully collected, resuspended in water, and centrifuged for 15 min at $75,000 \times g$ at 4 °C. The pellet was subjected to osmotic shock by resuspension in water for 15 min followed by centrifugation for 15 min at $12,000 \times g$ at 4 °C. After a second osmotic shock performed as above the pellet was resuspended in 0.35 M sucrose solution, added on top of 0.85 M sucrose solution, and centrifuged for 30 min at $75,000 \times g$ at 4 °C. The interphase containing purified myelin was carefully collected, resuspended in water, and centrifuged for 15 min at $75,000 \times g$. The supernatant was completely discarded, the pellet resuspended in 1× TBS (supplemented with protease inhibitor, Complete Mini, Roche), and stored at −80 °C. For lysate analysis, half brains of three Ctrl and iKO mice each at 8, 16, and 26 weeks pti were homogenized in modified RIPA buffer (1× TBS, 1 mM EDTA, 0.5% [w/v] Sodium deoxycholate, 1.0% [v/v] Triton X-100, cOmplete™ Mini protease inhibitor (Roche) using a T-10 basic Ultra-Turrax. Protein concentration was measured using the DC protein assay (BioRad Laboratories, Hercules, CA) according to the manufacturer's guidelines.

Immunoblotting was performed as previously described[65]. Purified myelin (0.5 µg for PLP/DM20; 2.5 µg for MBP) was separated on SDS-polyacrylamide gels (15% for PLP/DM20 and MBP) and blotted onto PVDF membranes (Hybond; Amersham) using the XCell II Blot Module (Invitrogen). Primary antibodies were incubated overnight at 4 °C in 5% milk in TBS with 0.1% Tween 20. HRP coupled secondary antibodies α-rabbit-HRP (Dako) or α-mouse-HRP (Dako) (1:10000) were incubated in 5% milk in TBS with 0.1% Tween 20 for 1 h at RT and detected using a CHEMOSTAR ECL & Fluorescence Imager (Intas). Quantification was performed in ImageJ (Fiji)[66] using actin or fast-green total protein as a loading control, graphs were plotted using GraphPad Prism 7.0. Statistical evaluation was performed using a two-tailed unpaired $t$ test (GraphPad Prism 7.0) per individual time point (iKO vs age-matched control). Levels of significance were displayed as $p < 0.05$ (*), $p < 0.01$ (**), and $p < 0.001$ (***).

**Proteome analysis of whole optic nerve lysates.** To investigate the systemic response to the loss of MBP and compact myelin in an adult animal we performed proteome analysis of whole optic nerves at 8, 16, and 40 weeks post tamoxifen injection. To compare the progressive loss of MBP during adulthood with the complete absence of MBP we also included optic nerves of 10 weeks old *shiverer* mice, a naturally occurring MBP knockout. Whole optic nerves were homogenized in 250 µl ice-cold modified RIPA buffer (1× TBS, 1 mM EDTA, 0.5%[w/v] sodium deoxycholate, 1.0%[v/v] Triton X-100, c0mplete™ Mini protease inhibitor (Roche Diagnostics)) using Teflon beads and a Precellys 24 tissue homogenizer (Bertin Instruments, France). Homogenization was carried out at a speed of 5500 r.p.m. for $2 \times 10$ s. A postnuclear supernatant was obtained by centrifugation with $1000 \times g$ at 4 °C to reduce undissolved tissue and cellular nuclei. The protein concentration of the supernatant was determined in triplicates using the BCA protein assay kit (Pierce). For quality control, 0.5 µg protein of each sample was loaded on a gel, and SDS-PAGE with subsequent silver staining of the gel was performed to visualize protein bands.

Supernatant fractions corresponding to 10 µg protein were subjected to in-solution digestion by filter-aided sample preparation (FASP) according to a protocol modified for processing of purified myelin as recently described in detail[64,67,68]. Optic nerve protein samples were lysed and reduced in lysis buffer (7 M urea, 2 M thiourea, 10 mM DTT, 0.1 M Tris pH 8.5) containing 1% ASB-14,

followed by dilution with ~10 volumes lysis buffer containing 2% CHAPS to reduce the ASB-14 concentration. Samples were loaded on centrifugal filter units (30 kDa MWCO, Merck Millipore), detergents were removed with wash buffer (8 M urea, 10 mM DTT, 0.1 Tris pH 8.5), proteins were alkylated with 50 mM iodoacetamide in 8 M urea, 0.1 M Tris pH 8.5, and excess reagent was removed with wash buffer. After buffer exchange with 50 mM ammonium bicarbonate (ABC) containing 10% acetonitrile, proteins were digested overnight at 37 °C with 400 ng trypsin in 40 μl of the same buffer. Recovered tryptic peptides were spiked with 10 fmol/μl of yeast enolase-1 tryptic digest standard (Waters Corporation) for quantification purposes and directly subjected to LC-MS analysis using nanoscale reversed-phase UPLC separation (nanoAcquity system, Waters Corporation) coupled to a quadrupole time-of-flight mass spectrometer with ion mobility option (Synapt G2-S, operated under MassLynx version 4.1, Waters Corporation). Peptides were trapped for 4 min at a flow rate of 8 μl/min 0.1% TFA on Symmetry C18 5 μm, 180 μm × 20 mm trap column and then separated at 45 °C on a HSS T3 C18 1.8 μm, 75 μm × 250 mm analytical column over 140 min at a flow rate of 300 nl/min with a gradient comprising two linear steps of 3–40% mobile phase B in 120 min and 40–60% mobile phase B in 20 min (A, water/0.1% formic acid; B, acetonitrile/0.1% formic acid). Mass spectrometric analysis was performed in the ion mobility-enhanced data-independent acquisition mode with drift time-specific collision energies (referred to as UDMS[E]) as introduced by Distler and colleagues[69,70] and adapted by us to synaptic protein fractions[71] and purified myelin[67,68]. For the correct quantification of exceptionally abundant proteins such as PLP and MBP, all samples were re-run in a data acquisition mode without ion mobility separation of peptides (referred to as MS[E]) to provide a maximal dynamic range at the cost of proteome coverage[67,68]. Continuum LC-MS data were processed for signal detection, peak picking, and isotope and charge state deconvolution using Waters ProteinLynx Global Server (PLGS) version 3.0.3. For protein identification, a custom database was compiled by adding the sequence information for yeast enolase 1 and porcine trypsin to the UniProtKB/Swiss-Prot mouse proteome (release 2018-11, 17001 entries) and by appending the reversed sequence of each entry to enable the determination of false discovery rate (FDR). Precursor and fragment ion mass tolerances were automatically determined by PLGS and were typically below 5 ppm for precursor ions and below 10 ppm (root mean square) for fragment ions. Carbamidomethylation of cysteine was specified as fixed and oxidation of methionine as variable modification. One missed trypsin cleavage was allowed. Minimal ion matching requirements were two fragments per peptide, five fragments per protein, and one peptide per protein. The FDR for protein identification was set to 1% threshold.

Per condition (8, 16, 40 weeks pti, shiverer), optic nerve fractions from four animals per genotype (Ctrl, iKO) were processed with replicate digestion, resulting in two technical replicates per biological replicate and thus in a total of 16 LC-MS runs to be compared. The mass spectrometry proteomics data have been deposited to the ProteomeXchange Consortium via the PRIDE[72] partner repository with the dataset identifier PXD025180.

**Antibodies.** Primary antibodies against the following antigens were used: APP 1:750 (Chemicon MAB348, RRID:AB_94882), CAII 1:1000 (gift from Said Ghandour[73]); CD3 1:150 (Abcam ab11089, RRID:AB_369097), GFAP 1:200 (Novocastra NCL-GFAP-GA5, RRID:AB_563739), Iba1 1:200 (Abcam ab5076, RRID:AB_2224402), MAC3 1:500 (Pharmingen 553322, RRID:AB_394780); MBP (this study: for generation of MBP antisera, rabbits were immunized with the intracellular peptide 105–115 of the 21.5 kDa isoform of mouse MBP (CQDENPVVHFFK). Anti-MBP antibodies were purified by affinity chromatography. The epitope is conserved in human and rat.), Olig2 1:100 (gift from Charles Stiles/John Alberta, DF308[74]), Caspr1 1:500 (Neuromab K65/35, RRID:AB_2083496), Nav 1.6 1:500 (Alomone ASC-009, RRID:AB_2040202), PDGFRA 1:500 (Cell Signaling 3174, RRID:AB_2162345), PLP1 1:500 (polyclonal rabbit, A431[75]), MAG 1:500 (Millipore Ab1567, RRID:AB_2137847), MOG 1:500 (gift from Christopher Linnington[76]), Actin 1:5000 (Millipore Mab 1501, RRID:AB_2223041).

Secondary antibodies for immunoblotting were HRP-goat anti-mouse IgG Dianova Cat# 115-035-003; 1:10,000; HRP-goat anti-rabbit, IgG Dianova Cat# 111-035-003; 1:10,000, HRP-goat anti-rat IgG Dianova Cat# 112-035-167; 1:10,000. Secondary antibodies for immunohistochemistry were donkey-α-Mouse Alexa 555, Invitrogen Cat# A31570, 1:1000; goat-α-rabbit Dylight 633, Invitrogen Cat# 35562, 1:500.

**Immunohistochemistry.** For Caspr1, Nav 1.6, and MBP labeling, mice were anesthetized with Avertin[77] and flushed with Hanks balanced salt solution followed by perfusion with 4% PFA in 0.1 M phosphate buffer. The brain and optic nerve were dissected. Optic nerves were postfixed in 4% PFA for 10 min and prepared for cryosectioning using a cryostat (Leica, Vienna, Austria). The brain was postfixed in 4% PFA for 24 h and prepared for paraffin embedding.

Slide-mounted optic nerve cryosections (9 μm) were air-dried at RT and washed three times 10 min in PBS followed by permeabilization for 1 h in 0.4% Triton-100 in PBS. Sections were blocked for 1 h in 1% fetal calf serum, 1% bovine serum albumin, and 1% fish skin gelatin in 1× PBS. The primary antibodies were then diluted in the blocking solution and incubated overnight at 4 °C. On the next day, slides were washed thrice with 1× PBS for 15 min each and incubated with the fluorescently labeled secondary antibody for 1 h at RT. After the incubation, the slides were washed thrice with 1× PBS for 15 min each, incubated with DAPI for 20 min, and washed again before being mounted using Aqua-Poly/Mount (Polysciences). Sections were stored at 4 °C until imaging by confocal fluorescence microscopy.

Brain hemispheres were embedded in paraffin, sectioned coronally, and labeled chromogenic for CD3, GFAP, APP, and MAC3 and with fluorescent antibodies for CAII, Olig2, and PDGFRa as essentially described in refs. [78,79]. TUNEL staining of paraffin fimbria sections for quantification of apoptotic cells was performed according to the manufacturer's protocols (DeadEnd™ Colorimetric TUNEL System, Promega). Imaging of 2 fimbria (GFAP, MAC3, APP, Olig2, CAII, PDGFRA) or 6 fimbria (CD3) per animal was performed on a bright-field light microscope (Zeiss AxioImager Z1 with Zeiss AxioCam MRc camera using the software ZEN 2012 blue edition) with the following magnifications: ×20 (CD3, GFAP, MAC3, Olig2, CAII, PDGFRA), ×40 (APP), and ×100 (representative images for display in the figures). Image analysis and quantification of markers were performed in Fiji[66] using a custom-made thresholding macro (GFAP, MAC3) or counted manually using the cell counter plugin (CD3, APP) in Fiji.

**EDU labeling.** Mice at the time point 40 weeks after tamoxifen administration received EdU in drinking water 0.2 mg/ml for 3 consecutive weeks. Mice were killed for analysis 3 weeks after the final EDU administration. Labeling of EDU positive cells on paraffin sections of fimbria was performed using the Click-iT™ Plus EdU Cell Proliferation Kit (Thermo Fisher Scientific) according to the manufacturer's protocol followed by fluorescent co-labeling for CAII, Olig2, or PDGFRA as described.

**Confocal fluorescence microscopy.** Confocal images of optic nerves immunolabeled for Caspr1 and NaV 1.6 were acquired on a SP5 confocal microscope (Leica Microsystems). Fluorescent signals were imaged sequentially to avoid cross bleeding using an HCX PL APO lambda blue 63.0 × 1.20 WATER UV objective. The following laser lines were used Argon Laser at 488 nm and 514 nm was used to excite Alexa 488 and Alexa 555 respectively. A HeliumNeon (HeNe) laser at 633 nm was used to excite Alexa 633 and Alexa 647. The confocal software LasAF was used for image acquisition. Images were saved as .lif files and quantified using the Fiji plugin cell counter. The number of nodes of Ranvier was quantified in 9 μm longitudinal cryostat sections of optic nerves 16 W and 40 W pti in at least 3 FOV at a size of 240 μm × 240 μm which were selected for maximal coverage with tissue. The nodes/area were normalized to the average of the controls.

**Electron microscopy.** Sample preparation of optic nerves by high-pressure freezing (HPF) and freeze substitution (FS) for transmission electron microscopy was performed as described[77]. Mice were killed by cervical dislocation and optic nerves were dissected and placed into an HPF specimen carrier with an indentation of 0.2 mm. The remaining volume was filled with 20% polyvinylpyrrolidone (Sigma-Aldrich, P2307-100 G) in PBS. Samples were cryo immobilized and fixed using a HPM100 (Leica) and freeze substituted using a Leica AFS (Leica Microsystems, Vienna, Austria) and embedded in Epon resin according to the protocol for optic nerves[77]. Tissues for conventional fixation used for quantification of corrected g-ratios and phenotype counting were dissected and immersion fixed for at least 24 h in 4% paraformaldehyde (PFA) and 2.5% glutaraldehyde in 0.1 M PB (containing 109.5 mM $NaH_2PO_4 \cdot H_2O$, 93.75 mM $Na_2HPO_4 \cdot 2H_2O$, and 86.2 mM NaCl), contrasted with osmium tetroxide, dehydrated and Epon embedded as described[77]. Ultrathin sections (50 nm) were cut using a UC-7 ultramicrotome (Leica Microsystems, Vienna, Austria) and contrasted with UranyLess™ (Science Services, Munich, Germany) for 30 min. Samples were imaged using a LEO 912AB Omega transmission electron microscope (Carl Zeiss, Oberkochen, Germany) with an on-axis 2048 × 2048 CCD camera (TRS, Moorenweis, Germany).

Normal appearing myelinated axons, axons with visible membrane tubulations, axonal degeneration, and unmyelinated axons were quantified on TEM micrographs. Quantification was performed on 3-4 animals per time point with at least 5 FOV with a total area of at least 330 μm² and at least 200 axons per animal. Statistical analysis was performed between iKO and the respective control using a two-tailed unpaired t test.

For quantification of inner tongue area and measurement of corrected g-ratio, images were analyzed using Fiji (https://imagej.net/Fiji). Axonal caliber (d) was calculated from the measured area (A) using Eq. (1):

$$d = 2\sqrt{\frac{A}{\pi}} \tag{1}$$

The inner tongue area was independently plotted as average per animal. Due to the occurrence of phenotype at the inner tongue, the area including phenotypical *shiverer*-like membranes and the axon ($A_n$) was subtracted using the following Eq. (2) to obtain the corrected fiber caliber $D_{corr}$:

$$D_{corr} = 2\sqrt{\frac{(A_m - A_n + A_a)}{\pi}} \tag{2}$$

with $A_m$: area of compact myelin, $A_n$: area of non-compact myelin and axon, and $A_a$:

axon area. The corrected $g$-ratio was then calculated as

$$g\text{-corr} = \frac{d}{D_{corr}} \tag{3}$$

Analysis was performed on at least 5 FOV with at least 150 axons per animal on 3–4 animals. For statistical analysis, we applied the Kolmogorow–Smirnow test. This test seeks differences between two datasets; it is non-parametric and distribution free. The null hypothesis of no difference between the datasets is rejected if $p$ is <0.05.

**Immunoelectron microscopy.** Immunogold labeling of cryosections prepared according to the Tokuyasu method was performed as previously described[77,80]. Optic nerves were dissected and immersion fixed in 4% PFA + 0.25% glutar-aldehyde in 0.1 M phosphate buffer overnight and cryo-protected using 2.3 M sucrose in 0.1 M phosphate buffer mounted onto aluminum pins for cryo-sectioning and frozen in liquid nitrogen. Ultrathin 50–80 nm cryosections sections were cut with a 35° diamond knife, cryo-immuno 2.0 mm (Diatome, Biel, Swit-zerland) using a Leica UC6 ultramicrotome with a FC6 cryochamber (Leica, Vienna, Austria). Primary antibodies used were specific for PLP (A431[75], and MBP (Custom-made MBP antibody, this study). Protein A-gold conjugates were obtained from the Cell Microscopy Center, Department of Cell Biology, UMC Utrecht, The Netherlands (https://www.cellbiology-utrecht.nl/products.html). Sections were imaged using a LEO EM912 Omega transmission electron micro-scope (Carl Zeiss Microscopy, Oberkochen, Germany). For quantification of MBP density at least 4 sections within a total of 300 $\mu m^2$ per animal were quantified using a 2 $\mu m$ grid to randomly select axons ($n = 3$–4 animals, optic nerve, 26 weeks pti). Gold particle number and compact myelin area per sheath were counted for every randomly selected axon using Fiji[66] and Microscopy image browser[81]. Graphs display gold particles per $\mu m^2$ compact myelin. Quantifications were per-formed blinded to the genotype and statistical analysis was performed using a two-tailed unpaired $t$ test in GraphPad prism 7.0 comparing control to iKO.

**Focused ion beam-scanning electron microscopy.** FIB-SEM was performed as described in[82,83]. To visualize the emergence of membrane structures along a myelinated internode we imaged optic nerves 16 weeks and 26 weeks after tamoxifen induction. Optic nerves were either prepared by HPF and FS as described above or fixed for 24 h in 4% paraformaldehyde (PFA) and 2.5% glu-taraldehyde in 0.1 M phosphate buffer. To achieve sufficient contrast for detection of backscattered electrons with the ESB detector, chemically fixed optic nerves were processed using a modified protocol of the reduced osmium-thiocarbohydrazide-osmium (rOTO) method[84] as described previously[85]. Nerves were transferred into embedding molds filled with Durcupan and polymerized at 60 °C for 48 h as previously described. Samples for FIB-SEM imaging of high-pressure frozen optic nerve were prepared as described above and embedded in Durcupan (Sigma-Aldrich) instead of Epon. Samples in blocks were then trimmed using a 90° diamond trimming knife (Diatome AG, Biel, Switzerland) and mounted on a SEM stub (Science Services GmbH, Pin 12.7 mm × 3.1 mm) using a silver-filled epoxy resin (Epoxy Conductive Adhesive, EPO-TEK EE 129–4; EMS) and polymerized at 60 °C overnight.

Optic nerves used for quantification of myelin coverage were minimal embedded in Durcupan as described[86,87] and polymerized at 60 °C for 48 h. Polymerized nerves were then also mounted on a SEM stub (Science Services GmbH, Pin 12.7 mm × 3.1 mm) using a silver-filled epoxy resin (Epoxy Conductive Adhesive, EPO-TEK EE 129–4; EMS) and polymerized at 60 °C overnight. All samples were coated with a 10 nm platinum layer using a sputter coater EM ACE600 (Leica) at 35 mA current. Samples were placed into the Crossbeam 540 focused ion beam-scanning electron microscope (Carl Zeiss Microscopy GmbH, Germany). To protect the sample surface, a 300–500 nm platinum or carbon layer was deposited on top of the region of interest. Atlas 3D (Atlas 5.1, Carl Zeiss Microscopy GmbH, Germany) software was used for milling and collection of 3D data. Initial milling was performed with a 15 nA current followed by a 7 nA current to polish the surface. Imaging was performed at 1.5 kV using an ESB detector (450 V ESB grid, pixel size $x/y$ 5 nm) in a continuous mill-and-acquire mode using 700 pA for the milling ($z$-step 50 nm).

Images were aligned using the ImageJ plugin TrackEM2[88] followed by postprocessing in Fiji: Images were cropped, inverted, and blurred (Gaussian blur, sigma 2) to suppress noise. Stacks were manually segmented using IMOD[89]. Quantification of phenotypes in stacks of optic nerves 16 and 26 W post tamoxifen was performed manually using Microscopy image browser[81]. Myelin coverage and the number of myelinoid bodies were measured in FIB-SEM stacks in $n = 3$ animals and stacks at 16 weeks pti with at least 90 axons and at 26 weeks pti in $n = 3$ animals with at least 100 axons per animal.

**Nano-SIMS imaging.** Semithin sections of Epon-embedded spinal cord samples were collected on finder grids (formvar and carbon-coated 200 square mesh copper grids, FCF200-1-Cu, Science Services, Munich, Germany) and regions of interest were mapped by taking images at increasing magnification by TEM. NanoSIMS imaging was performed as previously described[90,91] by a NanoSIMS 50L

(CAMECA, Gennevilliers Cedex, France) with an 8 kV Cesium primary source. To detect the presence of $^{12}C$ and $^{13}C$, the signals of $^{12}C^{14}N^-$ and $^{13}C^{14}N^-$ were measured. To reach the steady state of ionization, the samples were first implanted with a primary current of ~15 pA. A current of ~0.5–1 pA was applied during the imaging. Entrance slit and aperture slit were selected to obtain sufficient mass resolving power for a good separation of $^{13}C^{14}N^-$ peak from the interference peaks $^{12}C^{15}N^-$. The images were obtained with the raster size between 10 × 10 μm and 20 × 20 μm and 256 × 256 pixels, or the size >20 × 20 μm and 512 × 512 pixels.

Image exportation, layer addition, and drift correction were performed by the OpenMIMS plugin from Fiji[66] and self-written Matlab (The Mathworks Inc., Natick, MA) scripts were used for the analysis and correlation of EM and SIMS images. The pixels of secondary ion images produced by NanoSIMS represent the counts of the respective isotope. Then ROIs were manually defined based on the morphological structure in the correlation with TEM image of the same sample area. The isotopic $^{13}C^{14}N/^{12}C^{14}N$ ratio of the ROI was calculated for every pixel in the overlays of the $^{12}C^{14}N$ and $^{13}C^{14}N$ images. Then the average value across all pixels in the ROI was calculated and presented as a data point (% enrichment of $^{13}C$). The percentage of enrichment was calculated based on the difference compared to the $^{13}C^{14}N/^{12}C^{14}N$ ratio at the natural abundance of $^{12}C$ and $^{13}C$ (0.0112), which was set as 0% enrichment. The Matlab script used for this analysis is available upon request. The maps were presented in pseudo-color using the Fiji LUT Fire and Physics. In the EM image, the ROIs are indicated by numbers referring to the biological structure shown in the graph. We employed $t$ tests to determine whether the $^{13}C$ enrichment was significantly different between different biological structures.

**Quantification and statistical analysis.** The group size (number of animals = n) and the statistical test used are indicated in the respective figure legend. For cal-culation of SD and testing for significance MSExcel and Graphpad Prism were used.

**Analysis of proteomic data.** For label-free protein quantification, the freely available software ISOQuant version 1.6 (www.isoquant.net) was used for post-identification analysis including retention time alignment, exact mass and retention time (EMRT) and ion mobility clustering, peak intensity normalization, isoform/homology filtering, and calculation of absolute in-sample amounts for each detected protein[69,70,92]. Only peptides with a minimum length of seven amino acids that were identified with scores above or equal to 5.5 in at least two runs were considered. FDR for both peptides and proteins was set to a 1% threshold and only proteins reported by at least two peptides (one of which unique) were quantified using the TOP3 method[93]. The parts per million (ppm) abundance values (i.e. the relative amount (w/w) of each protein in respect to the sum over all detected proteins) were log2-transformed and normalized by subtraction of the median derived from all data points for the given protein. As described in detail recently[71], significant changes in protein abundance were detected by moderated $t$-statistics across all technical replicates using an empirical Bayes approach and false discovery (FDR)-based correction for multiple comparisons[94], realized in the Bioconductor R packages limma version 3.38.3 and $q$-value version 2.14.1. The relative abundance of a protein was accepted as altered for $q$-values < 0.05. To detect changes in normalized protein abundance over the course of MBP deficiency, we used limma to analyze the difference of differences with the interaction term (iKO40w-Ctrl40w)-(iKO8w-Ctrl8w) according to the limma User's Guide (https://bioconductor.org/packages/release/bioc/vignettes/limma/inst/doc/usersguide.pdf). The exact $q$-values are reported in Supplementary Data.

**Reporting summary.** Further information on research design is available in the Nature Research Reporting Summary linked to this article.

## Data availability

All datasets generated and/or analyzed during the current study are available from the corresponding author. The original 3D electron microscopy data (FIB-SEM image stacks) generated in this study on which the supplementary movies are based have been deposited at the EMPIAR repository (https://www.ebi.ac.uk/empiar/)[95] with the public accession codes EMPIAR-10906 (Supplementary Movie 2), EMPIAR-10907 (Supplementary Movie 3), and EMPIAR-10908 (Supplementary Movies 1 and 4). The mass spectrometry proteomics data have been deposited to the ProteomeXchange Consortium via the PRIDE[72] partner repository with the dataset identifier PXD025180. Source data are provided with this paper.

## Code availability

The Matlab (The Mathworks Inc., Natick, MA) scripts used for the analysis of NanoSIMS data can be obtained from S.O.R.

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

## Acknowledgements

We thank R. Jung, A. Fahrenholz, D. Hesse, and U. Kutzke for technical support; U. Suter for the inducible *Plp*-CreERT2 mice; and A. Leha for advice on proteome data analysis. We would like to thank Daniela Doda for helping with the investigation of remyelination in the optic nerve during her laboratory rotation in the MSc Neurosciences Program of the International Max Planck Research School. We are grateful for the competent and careful proofreading of the manuscript by J. Edgar. This work was funded through the German Research Foundation (DFG) (MO 1082/1-2, FOR2848, project 08 to W.M.) and DFG grants (WE 2720/2-2 and WE 2720/4-1 to H.B.W.) and by the Cluster of Excellence and DFG Research Center Nanoscale Microscopy and Molecular Physiology of the Brain, CNMPB (W.M., M.M., A.M.S., B.S., H.E., and K.-A.N) and by the SFB1286/B1 and VR (Swedish Research Council) to N.T.N.P. The study was further supported by the German Research Foundation (DFG) under Germany's Excellence Strategy - EXC 2067/1- 390729940 (W.M., S.R.O.), the SPP1757; TRR274-1 (H.E. and K.-A.N.) and by grant RI 1967/7-3 (S.O.R.). Further, K.-A.N is supported by the European Research Council (ERC 671048 "MyeliNano" advanced grant) and by the Dr. Miriam and Sheldon G. Adelson Medical Research Foundation.

## Author contributions

W.M., K.-A.N., and M.M. conceptualized the study and designed the experiments; M.M. conducted experiments and analyzed the data; A.M.S. and M.-T.W. provided 3D FIB-SEM data sets; K.K. conducted experiments and conceived the production of the antibody for detection of MBP, validated and purified it; O.J. and L.P. generated and analyzed the proteome data; P.A.-G. and N.T.N.P. conducted the NanoSIMS imaging, S.O.R. co-designed the SILAC study for NanoSIMs imaging and wrote the software tool (Matlab script) for the analysis of the data; T.R. and B.S. provided technical assistance in the processing of samples for electron microscopy; W.M. supervised the study and wrote the original draft; M.M., A.M.S., O.J., P.A.-G., N.T.N.P., S.O.R., H.B.W., H.E., and K.-A.N. reviewed and edited the draft; W.M., H.B.W., N.T.N.P., H.E., and K.-A.N acquired the funding of the study.

## Funding

## Competing interests

The authors declare no competing interests.
