## [Peer Review File · Nature Communications]

Reviewers' Comments:

Reviewer #1:

Remarks to the Author:

The study is designed to visualise turnover of the myelin protein MBP in existing myelin internodes. MBP synthesis was conditionally ablated in adult mice and ¹³C-lysine used to trace new protein synthesis. The manuscript is well written and clear, utilising a range of well described innovative techniques to show myelin turnover and abnormalities. The novelty of the approach and the important question being addressed will be of strong interest to myelin biologists. The findings may also be of relevance mechanistically for the field of multiple sclerosis research and for other demyelinating diseases.

The authors propose that 'the induced Mbp 368 knockout is gradually transforming the normal myelin sheath into shiverer myelin 369 tubules by integration of newly synthesized material.' The interpretation of the importance of newly synthesised myelin relies on the Nano SIMS data that is derived from one animal and a quantification and sampling method that is poorly described. The use of one animal for Nano SIMS analysis is not standard for the field e.g. doi:

10.1523/JNEUROSCI.1898-17.2018 and the applicability of findings for more than one animal needs to be addressed in terms of reproducibility of findings in line with discussions in <https://avs.scitation.org/doi/pdf/10.1116/1.4993628>.

The 3D EM analyses are conducted on only 1 or 2 animals in two experimental groups, despite an n of 3 per experimental group being more standard eg

<https://journals.plos.org/plosone/article?id=10.1371/journal.pone.0198131>; . Myelin thickness, outfoldings and optic nerve diameter vary considerably along the length of axons, even in normal optic nerve <https://www.nature.com/articles/s41598-018-22361-2> and in brain

<https://link.springer.com/article/10.1007/s00429-019-01844-6>. It does not appear that control mice were assessed in the 3D EM analyses showing internode shortening and juxtaparanodal myelin tabulation, so perhaps it would be useful for the authors to speculate on the likelihood of such features being visible in wt mice.

Low animal numbers are a feature of the study, with 3-4 animals per group in most experiments. This poses a problem as data are often somewhat scattered and apparent differences are often not statistically significant. For example, in FigS4B an apparent increase is not significant and is not noted, whereas in FigS3G' a similar scale increase is noted as not significant but perhaps indicating a mechanism of importance i.e. newly differentiated oligodendrocytes contributing to the increase in myelin gene transcripts. This study would be much improved with increased animal numbers allowing firm conclusions to be drawn one way or the other.

The choice of olig2 and PDGFRA to identify OPCs and oligodendrocyte numbers respectively (line 167) is not justified by literature or convention. Olig2 identifies oligodendroglia along the differentiation lineage and PDGFRA is expressed in OPCs and other cell types. Please adjust the descriptions of the cell types quantified and conclusions drawn accordingly. It may be that this description is an error as the markers are used correctly elsewhere in the manuscript.

The concept that decrease of MBP levels together with posttranslational modifications affecting membrane composition and thus somehow influencing levels of other myelin synthesis associated enzymes (line 510) is unclear and needs better explanation.

Additional points:

Fig S1; if trends are noted, individual p values should be provided.

Fig S3; x axis labels need revising to denote +/- iKO and move the olig 2 label to the y axis

Sampling strategy for quantification of immunohistochemistry outcomes in optic nerve (noted in the Methods) are not described and it is not clear what data in the Results are derived from this method. How many images were quantified per animal and from where in the nerve were the images sampled?

Melinda Fitzgerald

Reviewer #2:

Remarks to the Author:

The question addressed by this manuscript "how myelin function is affected by myelin turnover" represents a critical gap of knowledge and is truly of great importance in the field of neuroscience

because of its relevance to brain health and because of its several implications, not only to those interested in myelin biology but also to a much broader community, due to its implications for cognitive neuroscience and brain health.

Turnover implies the maintenance of a steady state equilibrium, while old components or molecules are degraded and replaced. Myelin turnover implies an equilibrium between myelin degradation and myelin synthesis. In this study, degradation is achieved by conditional genetic deletion of the first exon of the myelin basic protein gene (Mbp) in PLP expressing oligodendrocytes. This is of relevance as, with age, one would expect that oligodendrocytes generated in childhood would need to replace proteins to maintain proper function. The MBP protein has been very well studied in the field and recognized to serve a “zippering” function of myelin wraps, allowing for the tight appearance characteristic of compact myelin. Critical to this concept, has been the study of a mouse model with a naturally occurring mutation in the Mbp regulatory elements, called “shiverer”, due to its characteristic shivering phenotype.

Mice with the inducible Mbp null allele phenotypically resemble shiverer mice, which is often used as reference throughout the manuscript. There are several strengths of the paper, including the thorough characterization of the demyelination occurring in the Mbp null mice 8-16 weeks after recombination. Most of the beautiful data (in Figures 1-7) provide a convincing and detailed description of the events leading to myelin disruption, up to 26 weeks after interfering with MBP biosynthesis. The electron micrographs, obtained using focused ion beam scanning electron microscopy (FIB-SEM) and the characterization of nicely preserved samples after high pressure freezing provide a detailed identification of the events leading to demyelination, while the use of nanoscale secondary ion mass spectrometry (Nano-SIMS), in ¹³C lysine fed mice, allows for the identification of temporal patterns of protein levels over time. Perhaps not surprisingly, the morphological and biochemical patterns detected in the MBP null mice nicely resemble those detected in the shiverer mice, which are used as reference.

Unfortunately, while the “degradation” part component of the turnover is extremely well characterized, the “synthesis/regrowth” part is clearly underdeveloped. It is mentioned that the increased levels of Mbp mRNA occurring at 40-52 weeks post recombination (as shown in Fig. 1E) are likely contributed by new synthesis of MBP occurring in OPC, as new cells are detected at this time points (as shown in Supplemental Figures 3 and 4). The main problem, however, is that the biochemical data at 40 weeks as well as the histopathological data at this time point, reveal a prominent microglial and astrocytic response to demyelination.

This leads this reviewer asking whether the data provided provide an answer to the original question on the relationship between myelin function and turnover in physiological conditions, or simply provide a technologically superb characterization of a model of demyelination.

Based on these concerns, one is left asking whether this is the best model to study the functional consequences of myelin turnover. Without an extensive and thorough characterization of the new myelin formation at 52 weeks, the study appears to be focused on a model of myelin disorganization and degradation, followed by a typical response to injury.

A number of additional minor concerns are listed below:

1. Distinct areas are used for different assays: optic nerve, fimbria, corpus callosum, spinal cord. Presumably there are differences in each of this area. It would be helpful to systematically address the changes occurring in a given area and then compare and contrast with those occurring in a different region.
2. It is unclear why the time point at 52 weeks (when the authors claim that 83 percent of mRNA is provided by the newly generated OPC) is omitted by the proteomic data. This seems a critical data point to show recovery. The inclusion of baseline data would be important.
3. From figure 4A, it appears that the percentage of unmyelinated axons rather than decreasing, increases over time and is greater at 52 weeks than at 16 weeks. How do the authors interpret this? why is the new myelin formed by the newly generated oligodendrocytes not sufficient to rescue the loss of “old” myelin?
4. Figure legends need some work, specially legend for figures 2 (very hard to follow) and 3 (blue

color ?)

Reviewer #3:

Remarks to the Author:

In this manuscript, Meschkat et al. ablated MBP in young adult mice to prevent the formation of new myelin in the CNS. This led to progressive loss of compact myelin with shortening of internodes and loss of nodal, paranodal and juxtapanodal integrity. Since MBP is necessary for myelin compaction in the CNS, the loss of compact myelin and the development of a shiverer-like phenotype was used as a readout to evaluate the timing of myelin turnover. By using an isotope mapping approach, the authors identify the paranodal and juxtapanodal region at the inner tongue as the most likely location for integration of new myelin components into the existing myelin sheath. In parallel, the authors show increased occurrence of myelinoid bodies after ablation of MBP, which they identify as a mean of myelin disposal. This study is interesting, it utilizes advanced imaging techniques and a simple but convincing approach to show that the maintenance of the myelin sheath needs continuous integration of new myelin components and to evaluate the time required for the turnover of an internode. I think this model could however be further exploited to provide an evaluation of myelin turnover at different stages of adult life and to evaluate the timing of axonal degeneration after demyelination, which has surprisingly not been given much attention in this study. In addition, English writing including punctuation should be improved.

Specific comments:

1/ Please, revise the English and the punctuation to improve reading flow.

2/ The protocol of tamoxifen injections is not written in a clear way in the main text and is also not clear in Figure 1. Please, write it in a clearer way or remove it from the main text. In the material and methods, it is clearly explained.

3/ I recommend changing the name of the control mice to *Mbp^{fl/fl}*PlpCreERT2^{-/-}* or *Mbp^{fl/fl}*PlpCreERT2⁻* instead of *Mbp^{fl/fl}*PlpCreERT2^{wt}*. There is no wild type CreERT2, so this name is confusing.

4/ Were the control animals injected with tamoxifen? Control animals need to be injected with tamoxifen, but it is not clear from the text. Please, clarify.

5/ How are the levels of MBP protein at 40 and 52 weeks pti? Are they increased such as for *Mbp* mRNA levels?

6/ It is not necessary to give numerical values of figures in the main text, it renders the text difficult to read. In addition, method details do not need to be included in the main text.

7/ Ablation of MBP has been induced only at one time-point, at 2 months, which corresponds to very young adult age. It would be interesting to ablate MBP also at later time-points to know whether there is a time-point where myelin is not renewed anymore or where renewal speed slows down.

8/ It would also be interesting to determine when and whether axons suffer from the loss of myelin. Up to 52 weeks pti, it seems according to Fig. 4 that there is no increase of axon loss in the *Mbp* iKO optic nerve. If axons get remyelinated by non-recombined newly formed oligodendrocytes, a new round of tamoxifen injection could prevent remyelination and allow to determine when axon loss occurs after demyelination. How is it in other regions of the CNS? Are there regions where demyelinated axons are more sensitive to degradation?

Point-by-point response to the reviewers NCOMMS-20-24011-T

We would like to thank all the reviewers for their excellent comments, critical remarks and suggestions. We have conducted new experiments, modified the figures and added new data. These are presented in a point-by-point response below following each of the reviewer's comments (*italic*). We assigned in the edited manuscript a color to each of the reviewers to highlight the respective text changes. These colors are indicated in the response below.

Reviewer #1 (Remarks to the Author):

Point 1) *The study is designed to visualise turnover of the myelin protein MBP in existing myelin internodes. MBP synthesis was conditionally ablated in adult mice and ^{13}C -lysine used to trace new protein synthesis. The manuscript is well written and clear, utilising a range of well described innovative techniques to show myelin turnover and abnormalities. The novelty of the approach and the important question being addressed will be of strong interest to myelin biologists. The findings may also be of relevance mechanistically for the field of multiple sclerosis research and for other demyelinating diseases.*

The authors propose that 'the induced Mbp knockout is gradually transforming the normal myelin sheath into shiverer myelin tubules by integration of newly synthesized material.' The interpretation of the importance of newly synthesised myelin relies on the Nano SIMS data that is derived from one animal and a quantification and sampling method that is poorly described. The use of one animal for Nano SIMS analysis is not standard for the field e.g. doi: 10.1523/JNEUROSCI.1898-17.2018 and the applicability of findings for more than one animal needs to be addressed in terms of reproducibility of findings in line with discussions in

Text changes in response to reviewer #1 are made in grey.

Our response:

The NanoSIMS data highlight the integration of newly synthesized membranes into the existing myelin sheath. However, the approach using pulse-labeling of mice by feeding ^{13}C -lysine diet for several weeks is extraordinarily expensive, which allowed us to perform only a proof-of-principle study rather than a large cohort study for the original manuscript. In response to the reviewer's remark, however, we have now pulse labeled another animal for 60 days with the ^{13}C -lysine diet for NanoSIMS analysis. Indeed, we found a very similar ^{13}C enrichment in the annotated structures as originally reported, thereby confirming the original statements. The results added as Supplemental **Figure S7**. We would like to ask for the reviewer's understanding that the excessive cost of the ^{13}C -lysine diet for such an extended period prevents us from including even more mice to further confirm the results.

Upon this remark, and aiming at further clarification of the methods, we rewrote the text (line 282-292) the figure legend and the relevant part of the method section describing the quantification method (in blue).

This paragraph in the methods section now reads (line 1377):

"Image exportation, layer addition and drift correction, were performed by the OpenMIMS plugin from Fiji (Schindelin et al., 2012) and self-written Matlab (the Mathworks Inc, Natick, MA) scripts were used for the analysis and correlation of EM and SIMS images. The pixels of secondary ion images produced by NanoSIMS represent the counts of the respective isotope. Then ROIs were manually defined based on the morphological structure in the correlation with TEM image of the same sample area. The isotopic $^{13}\text{C}^{14}\text{N}/^{12}\text{C}^{14}\text{N}$ ratio of the ROI was calculated for every pixel in the overlays of the $^{12}\text{C}^{14}\text{N}$ and $^{13}\text{C}^{14}\text{N}$ images. Then the average value across all pixels in the ROI was calculated and presented as data point (% enrichment of ^{13}C). The % enrichment was calculated based on the difference compared to the $^{13}\text{C}^{14}\text{N} / ^{12}\text{C}^{14}\text{N}$ ratio at the natural abundance of ^{12}C and ^{13}C (0.0112), which was set as 0% enrichment. The Matlab script used for this analysis is available upon request. The maps were presented in pseudo-color using the Fiji LUT Fire and Physics. In the EM image, the ROIs are indicated

by numbers referring to the biological structure shown in the graph. We employed t-tests to determine if the ¹³C enrichment was significantly different between different biological structures.”

We would like to kindly ask the reviewer to let us know in case they are missing any particular methodological information here, for us to add further details if required.

New Figure S7:

Point 2) The 3D EM analyses are conducted on only 1 or 2 animals in two experimental groups, despite an n of 3 per experimental group being more standard eg <https://journals.plos.org/plosone/article?id=10.1371/journal.pone.0198131>; Myelin thickness, outfoldings and optic nerve diameter vary considerably along the length of axons, even in normal optic nerve <https://www.nature.com/articles/s41598-018-22361-2> and in brain <https://link.springer.com/article/10.1007/s00429-019-01844-6>. It does not appear that control mice were assessed in the 3D EM analyses showing internode shortening and juxtaparanodal myelin tabulation, so perhaps it would be useful for the authors to speculate on the likelihood of such features being visible in wt mice.

Our response:

The 3D analysis was indeed performed on both iKO and control mice represented in different colors in the respective graphs. We apologize if that was not sufficiently clearly stated in the original manuscript; the statement has been rephrased to better convey the information.

We have also followed the reviewers remark about the sample number for the 3D EM analysis. Here we acquired additional FIB-SEM data sets from 6 additional mice, now reaching a group size of n=3 for all experimental groups, i.e. for both control and iKO for two time points (16 weeks and 26 weeks after induction). All quantifications confirm and thus strengthen the original observations.

More specifically, we replaced the previous **Figure 5B** (showing individual axons as data points) by a new graph showing individual mice as data points (with controls in grey and iKO in orange). In the FIB-SEM data stack of each animal >90 axons were traced and the μm length covered by myelin was determined and expressed as % myelin coverage with fully myelinated at 100% coverage. We determined now a reduction by 27% (before: 25%) at 16 weeks and 60% (before 70%) at 26 weeks pti.

From the same datasets we also quantified the number of myelinoid bodies, now in a group size of n=3 mice. These data are shown in the new **Figure 7C**:

Here we expressed the number of myelinoid bodies per 10 μm axonal length. The data show that at 16 weeks after induction myelinoid bodies occur significantly more frequent in the iKO. At the later time point this difference is not significant anymore, probably due to the prevailing myelin loss in the iKO. These new results confirm and strengthen our conclusions.

Point 3) *Low animal numbers are a feature of the study, with 3-4 animals per group in most experiments. This poses a problem as data are often somewhat scattered and apparent differences are often not statistically significant. For example, in FigS4B an apparent increase is not significant and is not noted, whereas in FigS3G' a similar scale increase is noted as not significant but perhaps indicating a mechanism of importance i.e. newly differentiated oligodendrocytes contributing to the increase in myelin gene transcripts. This study would be much improved with increased animal numbers allowing firm conclusions to be drawn one way or the other.*

Our response:

We agree that it is generally desirable to have larger groups to obtain more significant results. However, we would like to ask for the reviewers understanding that apart from the NanoSIMS and 3D EM data in Figures S7, 5 and 7, we don't have additional samples available at this time. In particular, generating additional cohorts with tamoxifen injection and EdU application for the extended evaluation of oligodendrocyte death and OPC proliferation at high age would take over a year while all available evidence clearly indicates that widespread oligodendrocyte death does not occur in our model. The individual variability is probably caused by the complex experimental design with tamoxifen induction, ageing and EdU application. However, with the available data we can clearly exclude widespread oligodendrocyte death. In addition, qRT-PCR data shown in Figure S4C independently support our claim of the absence of major oligodendrocyte cell death.

Adding more mice would not only represent major effort and require over a year of time without much probability to affect our finding and conclusion. More importantly it would also require applying for a new animal license for breeding and investigating burdened animals, which is unlikely to be granted for the purpose of oligodendrocyte death given the results that were already gained. Together, we would like to ask for the reviewers understanding that we have not added more experiments to address oligodendrocyte death or numbers.

Point 4) *The choice of olig2 and PDGFRA to identify OPCs and oligodendrocyte numbers respectively (line 167) is not justified by literature or convention. Olig2 identifies oligodendroglia along the differentiation lineage and PDGFRA is expressed in OPCs and other cell types. Please adjust the descriptions of the cell types quantified and conclusions drawn accordingly. It may be that this description is an error as the markers are used correctly elsewhere in the manuscript.*

Our response:

We thank the reviewer for careful reading! This is indeed a mistake that escaped our attention. Thanks for making us aware of this! The text was changed accordingly.

Point 5) *The concept that decrease of MBP levels together with posttranslational modifications affecting membrane composition and thus somehow influencing levels of other myelin synthesis associated enzymes (line 510) is unclear and needs better explanation.*

Our response:

In response, for further clarification we rephrased the relevant paragraph, which now reads: "MBP interacts with negatively charged lipids such as PI(4,5)P2 and thereby influences lipid ordering and associates with galactosylceramid and cholesterol-rich lipid rafts in mature myelin (DeBruin et al., 2005, Fitzner et al., 2006, Musse et al., 2008, Debruin and Harauz, 2007, Ozgen et al., 2014). The decrease of MBP levels in the iKO together with changes in posttranslational modifications of the remaining MBP, e.g. deamination, could affect membrane ~~composition~~ organization and function (Boggs, 2006)."

Point 6) Additional points:

Fig S1; if trends are noted, individual p values should be provided.

Our response: As requested we added the p-values.

Fig S3; x axis labels need revising to denote +/- iKO and move the olig 2 label to the y axis

Our response: Thanks for the suggestion. We have changed the Figure accordingly.

Sampling strategy for quantification of immunohistochemistry outcomes in optic nerve (noted in the Methods) are not described and it is not clear what data in the Results are derived from this method. How many images were quantified per animal and from where in the nerve were the images sampled?

Our response: In response, we added the information about the sampling strategy in the methods section (line 1255-1258). Briefly, immunohistochemistry of optic nerves was performed for the quantification of nodes of Ranvier (Fig. 6C) by immunofluorescence labeling for Caspr1 and Nav1.6 and confocal microscopy. The results are described in line 392-395 of the manuscript and show a clear loss of the nodal organization in the iKO 40 weeks after induction. For quantification 3 fields of view (FOV) at a size of 240 μ m x 240 μ m were analyzed per nerve on longitudinal cryostat sections using the Fiji plugin cell counter and all detectable nodes were counted per FOV. Data points in Fig. 6C show individual mice with the number of nodes/area normalized to the average of the controls.

Reviewer #2 (Remarks to the Author):

The question addressed by this manuscript “how myelin function is affected by myelin turnover” represents a critical gap of knowledge and is truly of great importance in the field of neuroscience because of its relevance to brain health and because of its several implications, not only to those interested in myelin biology but also to a much broader community, due to its implications for cognitive neuroscience and brain health.

Turnover implies the maintenance of a steady state equilibrium, while old components or molecules are degraded and replaced. Myelin turnover implies an equilibrium between myelin degradation and myelin synthesis. In this study, degradation is achieved by conditional genetic deletion of the first exon of the myelin basic protein gene (Mbp) in PLP expressing oligodendrocytes. This is of relevance as, with age, one would expect that oligodendrocytes generated in childhood would need to replace proteins to maintain proper function. The MBP protein has been very well studied in the field and recognized to serve a “zippering” function of myelin wraps, allowing for the tight appearance characteristic of compact myelin. Critical to this concept, has been the study of a mouse model with a naturally occurring mutation in the Mbp regulatory elements, called “shiverer”, due to its characteristic shivering phenotype.

Mice with the inducible Mbp null allele phenotypically resemble shiverer mice, which is often used as reference throughout the manuscript. There are several strengths of the paper, including the thorough characterization of the demyelination occurring in the Mbp null mice 8-16 weeks after recombination. Most of the beautiful data (in Figures 1-7) provide a convincing and detailed description of the events leading to myelin disruption, up to 26 weeks after interfering with MBP biosynthesis. The electron micrographs, obtained using focused ion beam scanning electron microscopy (FIB-SEM) and the characterization of nicely preserved samples after high pressure freezing provide a detailed identification of the events leading to demyelination, while the use of nanoscale secondary ion mass spectrometry (Nano-SIMS), in ¹³C lysine fed mice, allows for the identification of temporal patterns of protein levels over time. Perhaps not surprisingly, the morphological and biochemical patterns detected in the MBP null mice nicely resemble those detected in the shiverer mice, which are used as reference.

Unfortunately, while the “degradation” part component of the turnover is extremely well characterized, the “synthesis/regrowth” part is clearly underdeveloped. It is mentioned that the increased levels of Mbp mRNA occurring at 40-52 weeks post recombination (as shown in Fig. 1E) are likely contributed by new synthesis of MBP occurring in OPC, as new cells are detected at this time points (as shown in Supplemental Figures 3 and 4). The main problem, however, is that the biochemical data at 40 weeks as well as the histopathological data at this time point, reveal a prominent microglial and astrocytic response to demyelination.

This leads this reviewer asking whether the data provided provide an answer to the original question on the relationship between myelin function and turnover in physiological conditions, or simply provide a technologically superb characterization of a model of demyelination.

Based on these concerns, one is left asking whether this is the best model to study the functional consequences of myelin turnover. Without an extensive and thorough characterization of the new myelin formation at 52 weeks, the study appears to be focused on a model of myelin disorganization and degradation, followed by a typical response to injury.

Text changes in response to reviewer #2 are marked in **light blue**.

Our answer:

We thank the reviewer for these critical remarks because this is a relevant point, in response to which we would like to clarify that our main interest is about – from the onset of the project on - the demyelination as a consequence of impaired myelin turnover.

The remyelination aspect at the late timepoint was observed rather as a side aspect since continuous maturation of OPCs to myelinating oligodendrocytes was shown before as a mechanism of myelin maintenance (Young et al (2013), Neuron 77: 873-885; doi: 10.1016/j.neuron.2013.01.006). In response to the request we added new data to the manuscript, we detected a large proportion of unmyelinated axons at the late time points, but also a small increase in the number of myelinated axons compared to 40 weeks pti. When we compared the proximal and distal part of the nerve, we found more myelinated axons in the proximal part which is a possible indication of remyelination by newly differentiated mature oligodendrocytes. Why this occurs only very late in our model might be explained by the lack of oligodendrocyte cell death. We added these new data as **Figure S8** to the manuscript and modified the text (line 332-339).

However, considering that we do not have additional high-age samples available at present, that we will not be able to gain them in due time, and that remyelination is not our intended main focus, we would like to ask for the reviewers' understanding that it is difficult for us to include further data on remyelination beyond that.

New Figure S8:

Non the less we would like to clarify that -in the assessment of the earlier timepoints- we were asking the question how the individual myelin sheath is renewed. In our model of *Mbp* ablation the site of incorporation of newly made membranes could be observed already at the time point of 16 weeks pti before demyelination and pathology fully developed in this mouse model. Only for this purpose it appeared indicated for us to show the recombined oligodendrocytes survive and continue to add new membrane to the myelin sheath.

A number of additional minor concerns are listed below:

1. Distinct areas are used for different assays: optic nerve, fimbria, corpus callosum, spinal cord. Presumably there are differences in each of this area. It would be helpful to systematically address the changes occurring in a given area and then compare and contrast with those occurring in a different region.

Our response: Assessing different CNS regions has mainly technical reasons. We used primarily the optic nerve to study the structural transition of the myelin sheath after depletion of MBP, since this white matter tract is best accessible for sample preparation for 3-dimensional electron microscopy (FIB-SEM). The reason to use spinal cord for NanoSIMS is the resolution limit of this method, which is not suited for the small diameter fibers in the optic nerve. Importantly, however, we found principally similar structural changes of the myelin sheath in all three regions (optic nerve, brain, spinal cord), notwithstanding that some differences in precise age at onset and progression may exist. The general CNS neuropathology was assessed in the brain, in which particularly the fimbria represents a well-myelinated model white matter tract, which according to our experience models very well the disease progression (see for example Lüders et al., 2019).

In response to the reviewers request we added further clarification in the manuscript (Line 297-301).

2. It is unclear why the time point at 52 weeks (when the authors claim that 83 percent of mRNA is provided by the newly generated OPC) is omitted by the proteomic data. This seems a critical data point to show recovery. The inclusion of baseline data would be important.

Our response: While we agree that an additional whole optic nerve proteome data set at 52 weeks could in principle provide further insight into the pathological state in this tissue at this stage, we feel that the considerable efforts required to generate and integrate such a data set are somewhat disproportional in view of the expected outcome, which is likely to be an indication of ongoing pathology and an increase in myelin proteins. Instead, we have addressed late abundance changes of myelin marker proteins in a targeted way by Western blotting, rather than by an unbiased proteomic approach, and have added the time points 40w and 52w to Figure 1F and Supplementary Figure 1D (see also below, our response to specific comment #5 of Reviewer 3).

As to data availability, we would like to note that we have deposited all mass spectrometry proteomics data to the ProteomeXchange/PRIDE repository. The data are accessible with Reviewer account details (username: reviewer_pxd025180@ebi.ac.uk; password: NJXmPdSw) and will be made publicly available in case of acceptance of this manuscript.

3. From figure 4A, it appears that the percentage of unmyelinated axons rather than decreasing, increases over time and is greater at 52 weeks than at 16 weeks. How do the authors interpret this? why is the new myelin formed by the newly generated oligodendrocytes not sufficient to rescue the loss of "old" myelin?

Our response: We would like to argue that the present model does not display massive OPC proliferation and differentiation, possibly owing to the lack of widespread oligodendrocyte death – in

difference to most other demyelinating mouse models. Please also see our response to comment #3 to reviewer #1.

4. Figure legends need some work, specially legend for figures 2 (very hard to follow) and 3 (blue color?)

Our response: We changed the figure legends for clarification. Kindly let us know in case you still see requirement for improvement here.

Reviewer #3 (Remarks to the Author):

In this manuscript, Meschkat et al. ablated MBP in young adult mice to prevent the formation of new myelin in the CNS. This led to progressive loss of compact myelin with shortening of internodes and loss of nodal, paranodal and juxtaparanodal integrity. Since MBP is necessary for myelin compaction in the CNS, the loss of compact myelin and the development of a shiverer-like phenotype was used as a readout to evaluate the timing of myelin turnover. By using an isotope mapping approach, the authors identify the paranodal and juxtaparanodal region at the inner tongue as the most likely location for integration of new myelin components into the existing myelin sheath. In parallel, the authors show increased occurrence of myelinoid bodies after ablation of MBP, which they identify as a mean of myelin disposal. This study is interesting, it utilizes advanced imaging techniques and a simple but convincing approach to show that the maintenance of the myelin sheath needs continuous integration of new myelin components and to evaluate the time required for the turnover of an internode. I think this model could however be further exploited to provide an evaluation of myelin turnover at different stages of adult life and to evaluate the timing of axonal degeneration after demyelination, which has surprisingly not been given much attention in this study. In addition, English writing including punctuation should be improved.

Specific comments:

1/ Please, revise the English and the punctuation to improve reading flow.

Text changes in response to reviewer #3 are made in **yellow**.

Our answer: In response, we have carefully re-checked language and punctuation with the help of a skilled colleague and native speaker. In case the reviewer still finds particular aspects that require improvement we would like to kindly ask them to be specified.

2/ The protocol of tamoxifen injections is not written in a clear way in the main text and is also not clear in Figure 1. Please, write it in a clearer way or remove it from the main text. In the material and methods, it is clearly explained.

Our answer: We simplified the text as requested.

*3/ I recommend changing the name of the control mice to Mbpfl/fl*PlpCreERT2^{-/-} or Mbpfl/fl*PlpCreERT2⁻ instead of Mbpfl/fl*PlpCreERT2^{wt}. There is no wild type CreERT2, so this name is confusing.*

Our answer: We agree with the reviewer and changed the nomenclature accordingly.

4/ Were the control animals injected with tamoxifen? Control animals need to be injected with tamoxifen, but it is not clear from the text. Please, clarify.

Our answer: Yes, all control animals were also treated with tamoxifen. We clarified this in the text.

5/ How are the levels of MBP protein at 40 and 52 weeks pti? Are they increased such as for Mbp mRNA levels?

Our answer: We thank the reviewer for this important question. We added the Western blot analysis of brain lysate at these time points for MBP, PLP, MAG and MOG and included them in Figure 1F and Supplementary Figure 1D and added the results to the manuscript in line 159-162.

Figure 1F

Figure S1 D:

6/ It is not necessary to give numerical values of figures in the main text, it renders the text difficult to read. In addition, method details do not need to be included in the main text.

Our answer: We thank for this recommendation and changed the text for clarity where appropriate.

7/ Ablation of MBP has been induced only at one time-point, at 2 months, which corresponds to very young adult age. It would be interesting to ablate MBP also at later time-points to know whether there is a time-point where myelin is not renewed anymore or where renewal speed slows down.

Our answer: We find that this question is of great importance. To investigate this, we have performed an experiment with animals treated with tamoxifen at the age of 6 months and analyzed the myelination state in the optic nerve at the age of 1 year (26 weeks pti) and added the new data as part of a revised Figure 4 (Figure 4B, see below). In brief, we see principally the same pathology as upon earlier induction, but with more normal-appearing myelinated axons. This may reflect that renewal dynamics changes with age, as suspected by the reviewer. This is now also mentioned in the manuscript in line 326-331.

8/ It would also be interesting to determine when and whether axons suffer from the loss of myelin. Up to 52 weeks pti, it seems according to Fig. 4 that there is no increase of axon loss in the Mbp iKO optic nerve. If axons get remyelinated by non-recombined newly formed oligodendrocytes, a new round of tamoxifen injection could prevent remyelination and allow to determine when axon loss occurs after demyelination. How is it in other regions of the CNS? Are there regions where demyelinated axons are more sensitive to degradation?

Our answer: Interestingly we found a number of axonal swellings as reflected in the APP immunohistochemistry, but no signs of actual axon loss. This agrees with prior findings in shiverer mice that also do not display axonal loss (Andrews et al. (2006) JNR 83:1533-1539; <https://doi.org/10.1002/jnr.20842>). A possible explanation might be that the transition from normally myelinated into a shiverer-like situation occurs so slowly that the axons can adapt to this. This phenomenon was also observed in a disease model: Edgar et al., EMBO Mol Med 2010 Feb;2(2):42-50. doi: 10.1002/emmm.200900057. We added text for clarification at line 322.

Reviewers' Comments:

Reviewer #1:

Remarks to the Author:

The authors have made substantial additions and improvements to the manuscript and have answered the questions I have raised.

Reviewer #2:

Remarks to the Author:

This Reviewer remains confused as to the modality of studying and defining myelin turnover. As previously mentioned the authors provide a number of convincing results that the late ablation of the first exon of MBP induces a demyelinating phenotype which resembled the dysmyelination reported for the shiverer mouse. Most of the manuscript provides evidence in support of this animal model: from the beautiful morphological data to the biochemical evidence and the authors clearly agree with this assessment (as reported on page 11, lines 254-256). The neuropathological data, consistent with the proteomic data further support reactive infiltration, microgliosis, astrogliosis (Fig 1 proteomic and Fig S6). In addition, there is clear evidence of remyelination, which is nicely shown by increased number of progenitors and new oligodendrocytes and by the presence of a greater number of newly myelinated axons at 52 weeks compared to earlier time points. Overall these results point to a nice model of demyelination followed by the formation of new myelin lacking MBP.

In addition, there are a number of confusing points. For instance, the lower levels of PLP at 8 weeks. If PLP half life is 6 months, how do the authors interpret its lower abundance at 8W pti? Why does MOG show a similar early decrease in abundance, while the time course of MAG is delayed? How do the authors reconcile these findings with their discussion of "metabolically active myelin" and the changes in the inner tongue?

Overall, based on the above, it remains hard to accept the described model as a way of studying physiological myelin turnover. As such, this manuscript appears to provide a very thorough characterization of a model of slow demyelination consequent to loss of MBP, in the absence of overt oligodendrocyte loss.

minor points:page 12 line 275 presumably there are few typos , as diet must have started prior to time of collection and analysis

Reviewer #3:

Remarks to the Author:

The authors have significantly improved the clarity of the manuscript and have appropriately answered my questions with additional experiments. I feel that the comments of the other reviewers were also appropriately answered. Therefore, I recommend publication of the revised manuscript in its current form.

Point-by-point response to the reviewers manuscript NCOMMS-20-24011B

We appreciate the effort and input of the reviewers. Here we address the remaining concerns point-by-point. Changes in the text of the manuscript are highlighted in yellow.

REVIEWER COMMENTS

Reviewer #1 (Remarks to the Author):

The authors have made substantial additions and improvements to the manuscript and have answered the questions I have raised.

We are very happy that we could address all questions.

Reviewer #2 (Remarks to the Author):

This Reviewer remains confused as to the modality of studying and defining myelin turnover. As previously mentioned the authors provide a number of convincing results that the late ablation of the first exon of MBP induces a demyelinating phenotype which resembled the dysmyelination reported for the shiverer mouse. Most of the manuscript provides evidence in support of this animal model: from the beautiful morphological data to the biochemical evidence and the authors clearly agree with this assessment (as reported on page 11, lines 254-256). The neuropathological data, consistent with the proteomic data further support reactive infiltration, microgliosis, astrogliosis (Fig 1 proteomic and Fig S6). In addition, there is clear evidence of remyelination, which is nicely shown by increased number of progenitors and new oligodendrocytes and by the presence of a greater number of newly myelinated axons at 52 weeks compared to earlier time points. Overall these results point to a nice model of demyelination followed by the formation of new myelin lacking MBP.

In addition, there are a number of confusing points. For instance, the lower levels of PLP at 8 weeks. If PLP half-life is 6 months, how do the authors interpret its lower abundance at 8W pti? Why does MOG show a similar early decrease in abundance, while the time course of MAG is delayed? How do the authors reconcile these findings with their discussion of "metabolically active myelin" and the changes in the inner tongue?

Our response:

In agreement with a remark by Reviewer #3 in point 4 of their 'further input' distinct myelin proteins are actually likely to exhibit differences in their turnover in our mouse model, depending on their specific localization in the myelin sheath. Thus, the relative abundance of PLP (in compact myelin), MOG (abaxonal) and MAG (an adhesion protein located on the adaxonal surface of the myelin sheath) are expected to be affected differently by the loss of compact myelin. To address this point, we have now included a respective statement into the discussion (page 18, line 431: "However, the observed changes in the abundance of other myelin proteins like PLP, MOG and MAG are likely due to the remodeling of the myelin sheath in this model rather than reflecting their normal half-lives.", highlighted in yellow).

The term "metabolically active myelin" in the discussion was cited from Hildebrandt et al. (1993). Our reason to adopt the term was to indicate that the concept of continuous myelin renewal by influx and disposal, e.g. in the form of myelinoid bodies, has already been proposed in the past. We have not modified our manuscript in this respect but we carefully considered how a modification could transmit this point better and are open to suggestions by the Reviewer or Editor. If specifically required by the Reviewer or Editor, we could also leave out the term "metabolically active myelin".

Overall, based on the above, it remains hard to accept the described model as a way of studying physiological myelin turnover. As such, this manuscript appears to provide a very thorough characterization of a model of slow demyelination consequent to loss of MBP, in the absence of overt oligodendrocyte loss.

Our response:

We agree with the Reviewer that the presented model has limitations if aiming at entirely truthfully reflecting physiological myelin turnover. Indeed, removal of MBP from myelin replenishment represents a major intervention in the system. Therefore, secondary neuropathology as a consequence of the induced demyelination was not unexpected.

However, our main focus was the ultrastructural visualization of the site of replenishment of compact myelin by *shiverer*-like myelin. The observed remodeling of the myelin sheath after MBP ablation tells us that the inner tongue is a critical sub-compartment of the myelin sheath with respect its maintenance and turnover. Furthermore, by using the degree of demyelination as a read-out, we were able to determine the time course of this replacement process in our model.

To improve our acknowledgement of the fact that our model is insofar non-physiological that the genetic change *per se* causes a modification of the structure of the myelin sheath and results in demyelination, we changed the abstract: Instead of the possibly misleading phrase “This raises the question how myelin function is affected by myelin turnover” we now phrase: “This raises the question how such a stable structure is renewed” and hope that this change is suited to address the reviewer’s concern.

Minor points: page 12 line 275 presumably there are few typos, as diet must have started prior to time of collection and analysis.

Our response:

We apologize for causing confusion. We were reporting the age of the mice along with the time point after induction with tamoxifen (weeks pti), which differ from each other as tamoxifen injection was not performed in newborn mice but in young adult mice 8 weeks of age. To better clarify this point we have changed the text (page 12, line 263, indicated in yellow) and omitted the age of the mice. Instead we now always give the time point after induction with tamoxifen (weeks pti).

Reviewer #3 (Remarks to the Author):

The authors have significantly improved the clarity of the manuscript and have appropriately answered my questions with additional experiments. I feel that the comments of the other reviewers were also appropriately answered. Therefore, I recommend publication of the revised manuscript in its current form.

Our answer:

We are glad that the Reviewer finds the revised manuscript is fit to be published.

Reviewer #3’s further input:

Our answer:

We agree with the Reviewer’s interpretations of our data as collected below and in the following just answer to those points that we understood as questions.

1- the work shows MBP is essential for myelination. Indeed, in the iKO study, formation of new compact myelin fails in the absence of MBP. Therefore, the work convincingly shows that the turnover of myelin and new compact myelin at the inner tongue require MBP.

2- part of the phenotype of the iKO mice may be due to the inability to replace MBP in the existing myelin, suggesting an increased breakdown of myelin. The latter could, to some extent, stimulate the formation of new myelin at the inner tongue.

3- the data also suggest that the remyelination observed at 52 weeks is due to new OPCs that were not recombined upon tamoxifen administration and express MBP as usual. The authors should consider to treat with tamoxifen again before remyelination occurs to recombine the new OPCs and test when axons start suffering (point 8 in reviewer 3's previous report). From their answer to point 8, it is not clear whether the authors have tested that or not.

Our answer:

We apologize that from our previous answer it was not entirely clear that we have not performed such an experiment. The reason is that we have focused more on the effects of the induced demyelination on the myelin sheaths rather than on the subsequent remyelination. However, we agree that it is in principal possible to perform such an experiment – though it would require about one and a half years of time and an additional animal license, which we estimate would take additional half a year. So, we would like to convince the Reviewer that considering our focus on the demyelination the possibility to perform such an experiment is an interesting possibility for future follow-up investigation.

4- concerning the data on the decreased expression of PLP at eight weeks pti (while its half-life has been shown to be six months), we can expect that the expression of myelin proteins such as PLP and MOG in the iKO mice will follow the breakdown of myelin in this model rather than their half-lives. Similarly, it is not surprising that MAG expression levels remain stable for longer time because MAG is not a compact myelin protein and localizes at the periaxonal membrane.

5- concerning reconciling the findings presented here and the discussion on 'metabolically active myelin', the data suggest that loss of myelin leads to more loss of PLP molecules than added in the myelin tubules at the inner tongue.

6- additionally, it's unclear how many mice per time point have been analysed in the experiments shown in Fig. 4E. G-ratio should be calculated per each mouse. The latter is the n that should be used for statistics.

Our answer:

For g-ratio analysis we used mostly n=3 and at one time point 4 iKO mice. Only subsequently we pooled the data of each group. For statistical analysis of the data shown in Figure 4E we applied the Kolmogorow-Smirnow test, which seeks differences between two datasets, is non-parametric and distribution free. The null hypothesis of no difference between the datasets is rejected if P is smaller than 0.05. This statistical test thus accounts for the fact that we are comparing multiple data points per group. We think that this test is more suitable to compare distributions and reflects better the complexity of g-ratio data including the individual variability than the t-test comparing g-ratio means between groups.

For clarification we added the numbers of animals to the figure legend and an explanation of the statistical test in the methods section (page 34, line 838).

For information we applied the t-test on the average g-ratio of the individual animals. As shown in the following graph, the time points 8 and 16 weeks pti show a trend and the last time point a significant difference to controls:

We are aware that the Kolmogorow-Smirnow test is not yet very common in the myelin community but it offers the above-mentioned advantages. If the reviewer favors the data representation in the above graph we can include this into the manuscript. However, it does not affect our conclusions.